# DHHC9-mediated GLUT1 S-palmitoylation promotes glioblastoma glycolysis and tumorigenesis

Zhenxing Zhang[1,6], Xin Li[1,2,6], Fan Yang[1,2,6], Chao Chen[1,6], Ping Liu[1], Yi Ren[1,2], Pengkai Sun[1,2], Zixiong Wang[1,2], Yongping You[3,4], Yi-Xin Zeng [1,5] & Xinjian Li [1,2✉]

Glucose transporter GLUT1 is a transmembrane protein responsible for the uptake of glucose into the cells of many tissues through facilitative diffusion. Plasma membrane (PM) localization is essential for glucose uptake by GLUT1. However, the mechanism underlying GLUT1 PM localization remains enigmatic. We find that GLUT1 is palmitoylated at Cys207, and S-palmitoylation is required for maintaining GLUT1 PM localization. Furthermore, we identify DHHC9 as the palmitoyl transferase responsible for this critical posttranslational modification. Knockout of DHHC9 or mutation of GLUT1 Cys207 to serine abrogates palmitoylation and PM distribution of GLUT1, and impairs glycolysis, cell proliferation, and glioblastoma (GBM) tumorigenesis. In addition, DHHC9 expression positively correlates with GLUT1 PM localization in GBM specimens and indicates a poor prognosis in GBM patients. These findings underscore that DHHC9-mediated GLUT1 S-palmitoylation is critical for glucose supply during GBM tumorigenesis.

---

[1] CAS Key Laboratory of Infection and Immunity, CAS Center for Excellence in Biomacromolecules, Institute of Biophysics, Chinese Academy of Sciences, 100101 Beijing, China. [2] College of Life Sciences, University of Chinese Academy of Sciences, 100049 Beijing, China. [3] Department of Neurosurgery, The First Affiliated Hospital of Nanjing Medical University, 210029 Nanjing, China. [4] Institute for Brain Tumors, Jiangsu Key Lab of Cancer Biomarkers, Prevention and Treatment, Jiangsu Collaborative Innovation Center for Cancer Personalized Medicine, Nanjing Medical University, 211166 Nanjing, China. [5] State Key Laboratory of Oncology in South China; Collaborative Innovation Center for Cancer Medicine, Sun Yat-Sen University Cancer Center, 510060 Guangzhou, China. [6] These authors contributed equally: Zhenxing Zhang, Xin Li, Fan Yang, Chao Chen. ✉email: lixinjian@ibp.ac.cn

GLUT1, encoded by *SLC2A1*, is a transporter facilitating the uptake of glucose in many tissues and belongs to the facilitative glucose transporter family that consists of 12 members[1]. To support rapid proliferation, cancer cells take up more glucose for glycolysis even in the presence of oxygen, a phenomenon known as the Warburg effect[2]. Elevated expression levels of GLUT1 were observed in many malignant tumors and correlated with poor clinical outcomes in patients[3–6]. Despite its upregulation in many types of human cancer, the mechanisms underlying GLUT1-promoted tumor malignant progression remain largely unknown.

A previous study on the palmitoylation of the blood–brain barrier capillary proteins has shown that GLUT1 is palmitoylated[7]. Palmitoylation of cysteine residues, the major modality of thioacylation on cellular proteins by the 16-carbon fatty acid palmitate, is catalyzed by palmitoyl acyltransferases (PATs)[8]. Over 800 putative palmitoylated proteins have been identified in mouse adipose tissues and adipocytes[9]. Human PATs contain a conserved zinc finger DHHC motif (Asp–His–His–Cys) within a cysteine-rich domain (CRD), thus belong to the zinc finger DHHC domain-containing protein family, which comprises 23 enzymes named DHHC1 to DHHC24 (without DHHC10)[10]. The hydrophobic palmitate moiety of palmitoylated proteins serves as a lipid anchor to facilitate the interaction between proteins and membranes that is critical for subcellular trafficking and membrane localization of intracellular proteins[10–12]. For example, H-RAS and N-RAS are palmitoylated by DHHC9/GCP16 PAT complex, leading to plasma membrane (PM) localization of these proteins[13,14]. Loss-of-function mutations in DHHC9 have been identified in patients with X-linked intellectual disability (XLID) and associate with an increased epilepsy risk[15–17]. In addition, DHHC9-mediated palmitoylation maintains protein stability and cell surface distribution of PD-L1, leading to immune escape of breast cancer cells[18].

In this study, we demonstrate that GLUT1 palmitoylation occurs on GBM cell lines and plays an important role in GLUT1 PM localization. Further mammalian PATs screening assay show that DHHC9 is the dominant palmitoyltransferase for GLUT1. DHHC9 palmitoylates GLUT1 at Cys207 to maintain PM localization of GLUT1, leading to a high level of glycolysis, thereby promoting GBM tumorigenesis.

## Results

**S-palmitoylation maintains PM localization of GLUT1.** To determine whether subcellular localization of GLUT1, the widely expressed glucose transporter responsible for the constant uptake of glucose[19], is regulated by S-palmitoylation, we examined its cellular distribution upon treatment with 2-bromopalmitate (2-BP), a pan-inhibitor of PATs. Immunofluorescence (IF) analyses of U87 and T98G human GBM cells with an anti-GLUT1 antibody and co-staining the cells with cholera toxin subunit B (CTB) conjugated with Alexa Fluor 594, a tracer for the PM, showed that 2-BP treatment induced redistribution of GLUT1 protein from the PM to the cytoplasm (Fig. 1a). Similarly, PM localization of GLUT1, as evidenced by increased cytoplasmic localization of eGFP-GLUT1 in the cells, was inhibited by 2-BP treatment (Supplementary Fig. 1a). Cell fractionation analyses confirmed that treatment of U87 and T98G cells with 2-BP substantially reduced the levels of endogenous GLUT1 associated with the PM fractions but did not affect total GLUT1 protein levels (Fig. 1b).

To determine whether GLUT1 is palmitoylated, we performed a metabolic incorporation assay using a bioorthogonal fatty acid analog (alkynyl palmitic acid) with click chemistry conjugation. Endogenous GLUT1, but not another glucose transporter GLUT3 in GBM cells, is S-palmitoylated through thioester bonds that are cleavable by treatment with hydroxylamine (HAM) (Fig. 1c and Supplementary Fig. 1b). To quantify the palmitoylation levels of GLUT1, we performed an acyl-PEG exchange (APE) assay, which enables the separation and semi-quantification of palmitoylated protein. Endogenous GLUT1 was highly palmitoylated (~90%), and its palmitoylation levels were dramatically reduced upon treatment with 2-BP (Fig. 1d). Omission of HAM treatment abolished the palmitoylation of GLUT1, which confirmed that GLUT1 is S-palmitoylated through thioester bonds (Fig. 1d). Moreover, GLUT1 can be labeled efficiently by alkynyl palmitic acid, alkynyl myristic acid, and alkynyl stearic acid probes, but much less efficiently labeled by an alkynyl arachidonic acid probe (Supplementary Fig. 1c), suggesting that palmitoylation, myristoylation, and stearoylation are the major acylation forms of GLUT1. In addition, results from pulse-chase experiments showed that palmitoylation of GLUT1 is a dynamic process with an approximate turnover half-life of 1 h (Supplementary Fig. 1d). Together, these findings indicate that GLUT1 is palmitoylated and that this post-translational modification is required for maintaining GLUT1 PM localization.

**GLUT1 is palmitoylated at Cys207 and S-palmitoylation is required for maintaining GLUT1 PM localization.** To determine the site of GLUT1 palmitoylation, we mutated every cysteine to serine in the GLUT1 protein, and their S-palmitoylation levels were characterized by using the alkynyl palmitic acid incorporation assay. The GLUT1 C207S mutation abolished palmitoylation without altering its protein expression level (Fig. 2a), which suggests that C207 is required for GLUT1 palmitoylation. Notably, this palmitoylation site at C207 of GLUT1 is located in the intracellular loop adjacent to the PM (Supplementary Fig. 2a) and is highly conserved among different species (Supplementary Fig. 2b). Similar to the 2-BP treatment, the eGFP-GLUT1 C207S mutant displayed cytoplasmic distribution in U87 and T98G cells (Supplementary Fig. 2c).

In line with this finding, reconstituted expression of Flag-tagged sgRNA-resistant (r) GLUT1 C207S in endogenous GLUT1-knockout (KO) U87 and T98G cells markedly blocked the incorporation of alkynyl palmitic acid onto GLUT1 (Fig. 2b, c). These results were validated by using the APE assays, which showed that palmitoylation levels of GLUT1 were dramatically reduced by reconstituted expression of rGLUT1 C207S as compared with wild-type (WT) rGLUT1 (Fig. 2d). Similar to that observed in 2-BP-treated U87 and T98G cells (Fig. 1a, b), the rGLUT1 C207S-variant protein showed a cytoplasmic distribution (Fig. 2e) and was depleted from the PM fractions of U87 and T98G cells (Fig. 2f).

**DHHC9 mediates GLUT1 S-palmitoylation and maintains GLUT1 PM localization.** To identify the potential palmitoyl acyltransferases that regulate GLUT1, a set of lentiviral plasmids expressing guide RNAs (gRNAs) targeting PATs were constructed and used in a clustered regularly interspaced short palindromic repeats (CRISPR) screen. Expression of DHHC9 gRNA abolished the incorporation of alkynyl palmitic acid onto GLUT1 (Supplementary Fig. 3a), suggesting that DHHC9 is one of the PATs that regulate GLUT1 palmitoylation in GBM cells. In reconfirmation assays, depleting DHHC9 with short hairpin RNAs (shRNAs) abrogated GLUT1 palmitoylation (Supplementary Fig. 3b). Co-immunoprecipitation (Co-IP) analyses showed that DHHC9 and GLUT1 form a complex in U87 and T98 cells (Supplementary Fig. 3c). Furthermore, purified DHHC9 directly bound to purified GLUT1 in an in vitro binding assay (Supplementary Fig. 3d). In addition, IF analyses with specificity-validated antibodies (Supplementary Fig. 3e) demonstrated that endogenous DHHC9 and GLUT1 colocalized at the PM of

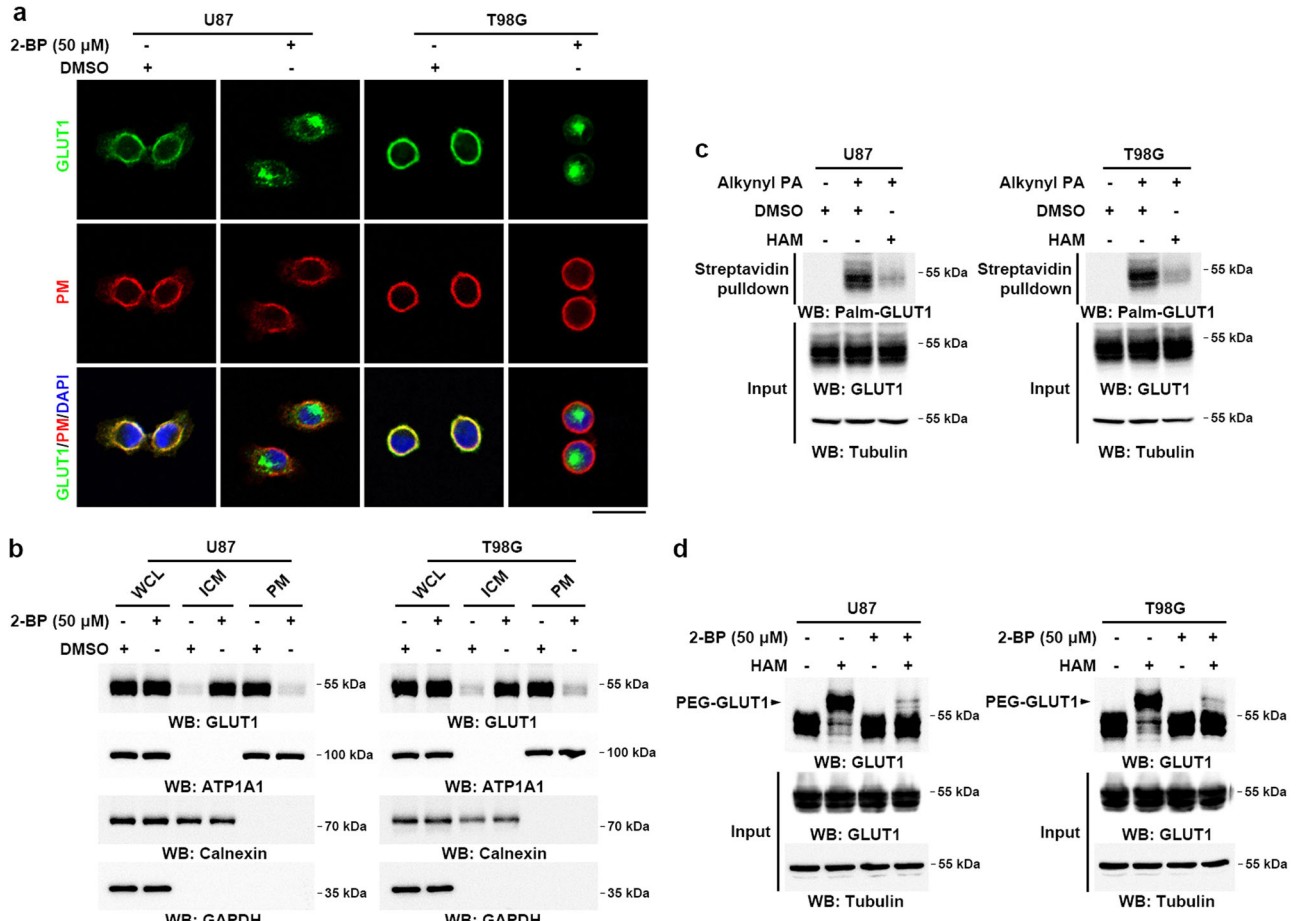

**Fig. 1 S-palmitoylation is required for maintaining GLUT1 PM localization. a** U87 or T98G cells were treated with DMSO or 50 μM of 2-BP for 8 h followed by incubation with 1 μg ml$^{-1}$ of cholera toxin subunit B (CTB) conjugated with Alexa Fluor 594 for 5 min at 37 °C. Endogenous GLUT1 cellular localization was visualized by immunofluorescent staining using antibodies against GLUT1. PM plasma membrane. Scale bar, 20 μm. **b** U87 or T98G cells were treated with DMSO or 50 μM of 2-BP for 8 h. Levels of GLUT1 in PM and ICM fractions were analyzed by immunoblotting. ATP1A1, calnexin, and GAPDH served as the marker of PM, ICM, and cytosol fraction, respectively. PM plasma membrane, ICM intracellular membrane, WCL whole-cell lysate. **c** GLUT1 palmitoylation was analyzed in lysates derived from U87 or T98G cells metabolically labeled with a palmitoylation probe (50 μM alkynyl palmitic acid [PA]) for 4 h by click reaction and streptavidin bead pulldown in the absence or presence of hydroxylamine (HAM), followed by immunoblotting using indicated antibodies. **d** GLUT1 palmitoylation levels in U87 or T98G cells were analyzed by the APE assays, upon 50 μM of 2-BP treatment in the absence or presence of HAM. PEG-GLUT1 bands indicated palmitoylated GLUT1. Representative results were obtained from at least three independent experiments with similar results. Source data are provided as a Source Data file.

U87 and T98 cells (Supplementary Fig. 3f). Consistently, robust reconstituted GFP fluorescence was observed at the PM of U87 and T98 cells in a split-GFP system[20] coexpressing C-terminal split-GFP (S1-10)-tagged DHHC9 and split-GFP (S11)-tagged GLUT1 (Supplementary Fig. 3g, h), demonstrating that DHHC9 and GLUT1 colocalize at the PM of these cells. Taken together, these data suggest that DHHC9 interacts and palmitoylates GLUT1 at the PM of GBM cells.

PAT-catalyzed S-palmitoylation of proteins utilizes palmitoyl-CoA as the palmitate donor. Thus, we performed an in vitro palmitoylation assay by mixing highly purified recombinant DHHC9 and GCP16, which have been reported to constitute a PAT[13,14], with GLUT1 in the presence of palmitoyl alkyne-CoA as the palmitate donor (Fig. 3a). The terminal alkyne group of this compound allows accessing the palmitoylation of proteins by use of the click chemistry linking reactions. Consistent with this observation, WT DHHC9 in complex with GCP16, but not single WT DHHC9 or catalytically inactive DHHC9 C169S (Supplementary Fig. 4a) in complex with GCP16, was able to incorporate palmitoyl alkyne (Supplementary Fig. 4b). Importantly, WT GLUT1, but not GLUT1 C207S, was palmitoylated by WT

DHHC9 in complex with GCP16, but not DHHC9 C169S in complex with GCP16 (Fig. 3a). Similar results were also observed in an in vitro palmitoylation assay using palmitoyl-CoA as the palmitate donor and by detecting GLUT1 and DHHC9 palmitoylation in the APE assays (Fig. 3b).

To further validate our findings in GBM cells, we used CRISPR/Cas9 genome-editing technology to knockout endogenous DHHC9 and reconstitutively expressed Flag-tagged WT rDHHC9 or rDHHC9 C169S mutant (Fig. 3c). Endogenous DHHC9 KO substantially blocked GLUT1 palmitoylation (Fig. 3d, e). These effects were abrogated by reconstituted expression of WT rDHHC9, but not the catalytically inactive rDHHC9 C169S mutant (Fig. 3d, e). As expected, endogenous DHHC9 KO rendered GLUT1 (Fig. 3f) and eGFP-GLUT1 largely cytoplasmic (Supplementary Fig. 4c), resembling the distribution observed after 2-BP treatment (Fig. 1a), and this effect was abrogated by reconstituted expression of WT rDHHC9, but not rDHHC9 C169S (Fig. 3f and Supplementary Fig. 4c). Consistently, cellular fractionations of U87 and T98G cells confirmed that DHHC9 KO substantially suppressed GLUT1 PM association (Fig. 3g), and this suppression was abrogated by reconstituted expression of WT

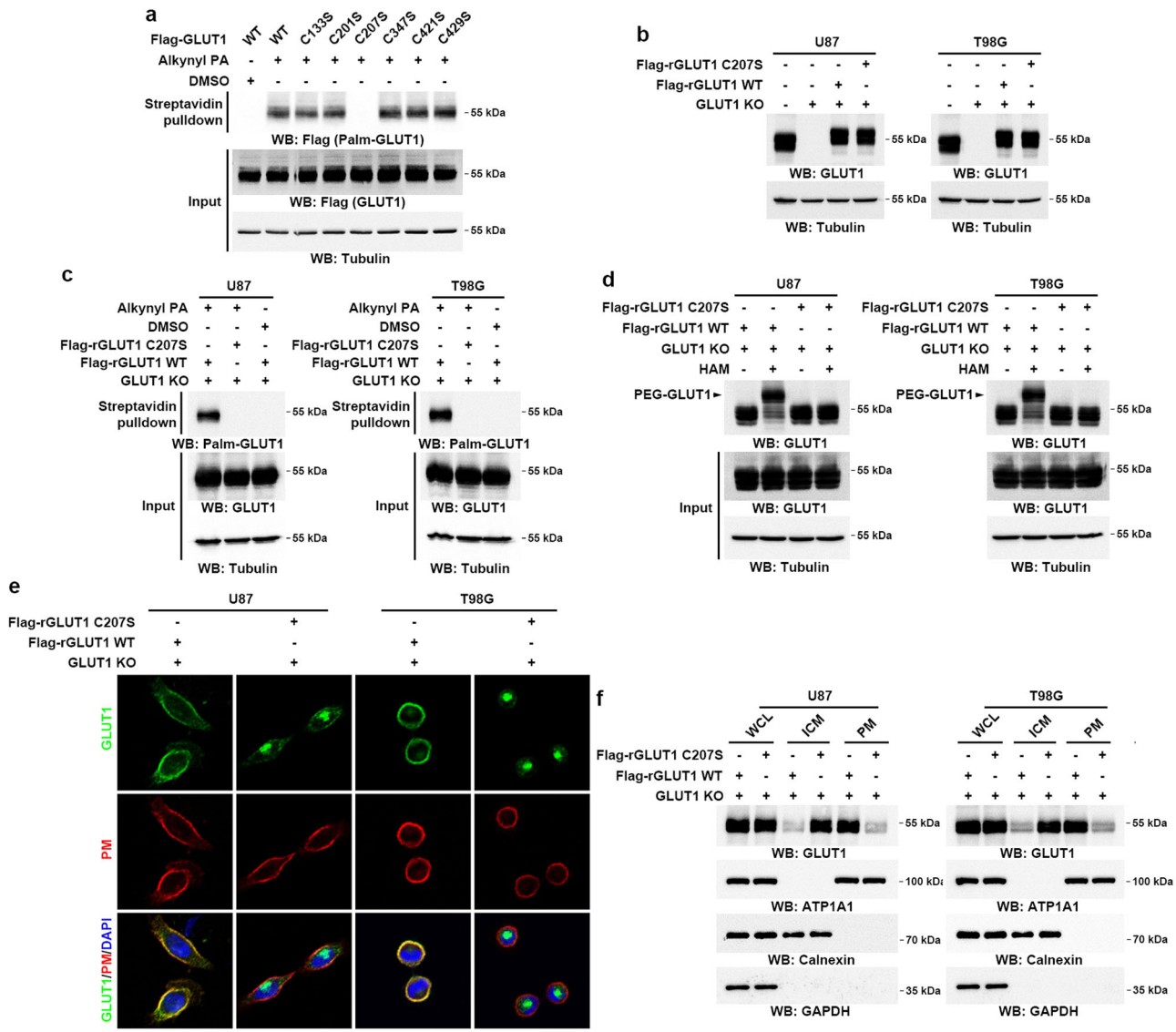

**Fig. 2 Cys207 palmitoylation is required for maintaining GLUT1 PM localization. a** U87 cells were transfected with Flag-tagged wild-type (WT) GLUT1 or indicated mutants. Twenty-four hours after transfection, the cells were metabolically labeled with 50 μM alkynyl PA for 4 h. Flag-GLUT1 palmitoylation levels were analyzed by click reaction and streptavidin bead pulldown, followed by immunoblotting. **b** Flag-tagged WT rGLUT1 or rGLUT1 C207S was reconstitutively expressed in U87 or T98G cells with the knockout of endogenous GLUT1. Immunoblotting was performed using indicated antibodies. **c** U87 or T98G cells with reconstituted expression of Flag-tagged WT rGLUT1 or rGLUT1 C207S were metabolically labeled with 50 μM alkynyl PA for 4 h. Palmitoylation levels of Flag-rGLUT1 were analyzed by click reaction and streptavidin bead pulldown, followed by immunoblotting. **d** APE assay was performed to analyze the GLUT1 palmitoylation in U87 or T98G cells with reconstituted expression of Flag-tagged WT rGLUT1 or rGLUT1 C207S. The top band indicates the palmitoylated GLUT1 (PEG-GLUT1). **e** U87 or T98G cells with reconstituted expression of Flag-tagged WT rGLUT1 or rGLUT1 C207S were incubated with 1 μg ml⁻¹ of CTB conjugated with Alexa Fluor 594 for 5 min at 37 °C. GLUT1 cellular localization was visualized by immunofluorescent staining using an antibody against GLUT1; and the plasma membrane was marked by Alexa Fluor 594-conjugated CTB. PM, plasma membrane. Scale bar, 20 μm. **f** Levels of GLUT1 in PM and ICM fractions were analyzed by immunoblotting in U87 or T98G cells with reconstituted expression of Flag-tagged WT rGLUT1 or rGLUT1 C207S. ATP1A1, calnexin, and GAPDH served as the marker of PM, ICM, and cytosol fraction, respectively. PM plasma membrane, ICM intracellular membrane, WCL whole-cell lysate. Representative results were obtained from at least three independent experiments with similar results. Source data are provided as a Source Data file.

rDHHC9, but not rDHHC9 C169S. Similarly, reducing the expression of DHHC9 by adenovirus-mediated shRNAs significantly suppressed GLUT1 PM localization in PDX cells (Supplementary Fig. 4d–f). Collectively, these data suggest that DHHC9 palmitoylates GLUT1 to maintain its PM localization.

We next investigated whether DHHC9-mediated GLUT1 S-palmitoylation regulates GLUT1 plasma membrane localization in normal astrocytes. Notably, DHHC9 and GLUT1 are highly expressed in GBM cells compared with non-transformed normal

human astrocytes (NHAs) (Supplementary Fig. 5a). Endogenous DHHC9 depletion by adenovirus-mediated shRNAs markedly blocked palmitoylation (Supplementary Fig. 5b, c) and PM association (Supplementary Fig. 5d) of GLUT1 in NHAs. These results suggest that DHHC9-mediated GLUT1 palmitoylation is also required for maintaining GLUT1 PM localization in normal human astrocytes.

Given that PM localization of GLUT1 depends on protein kinase B (PKB, also known as AKT) and Protein kinase C (PKC)

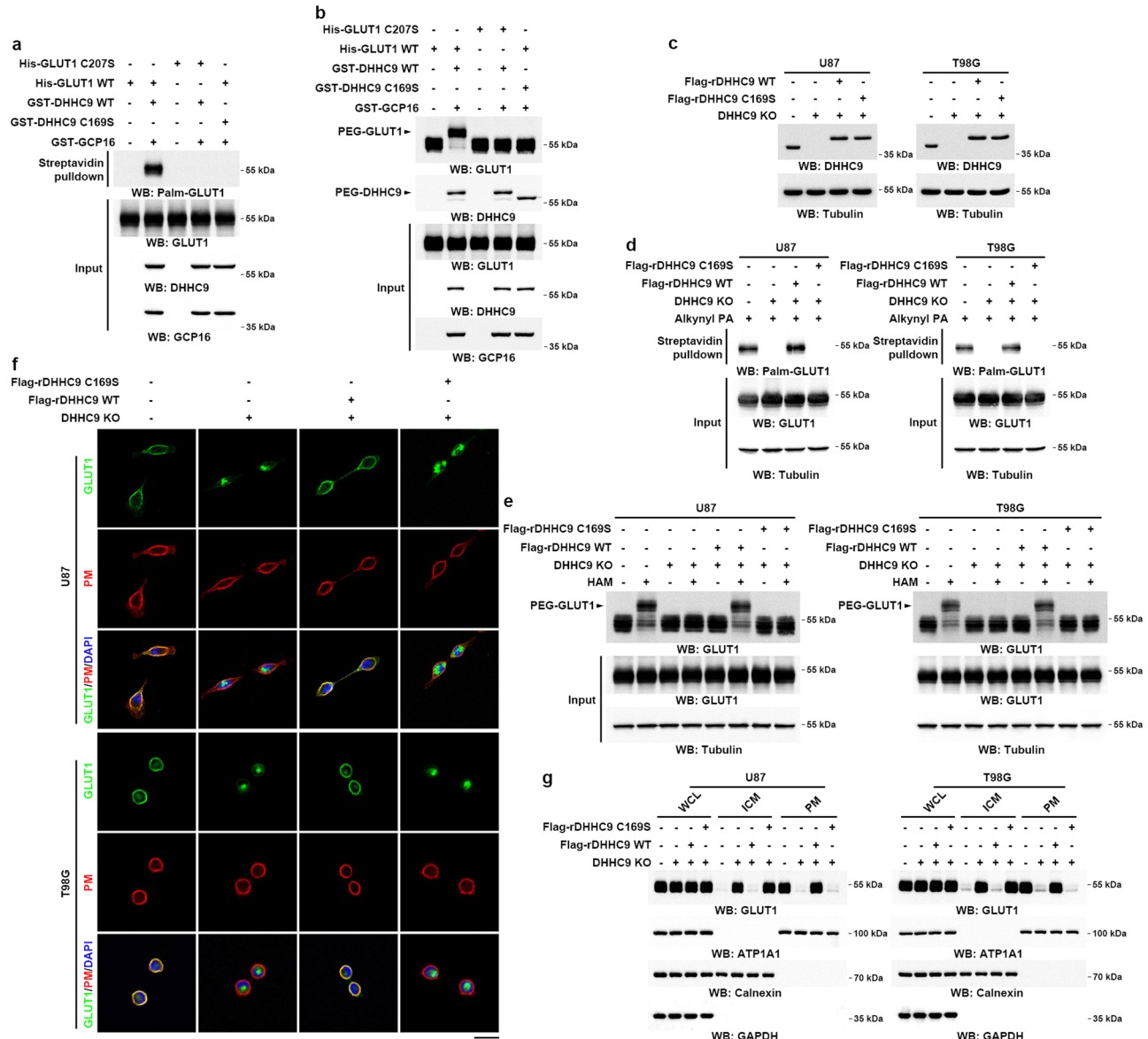

**Fig. 3 DHHC9 palmitoylates GLUT1 at Cys207 to maintain GLUT1 PM localization. a** In vitro palmitoylation analysis was performed by mixing purified WT DHHC9/GCP16 and DHHC9 C169S/GCP16 with purified WT GLUT1 or GLUT1 C207S in the presence of palmitoyl alkyne-CoA. GLUT1 palmitoylation levels were analyzed by click reaction and streptavidin bead pulldown, followed by immunoblotting. **b** In vitro palmitoylation analysis was performed by mixing purified WT DHHC9/GCP16 and DHHC9 C169S/GCP16 with purified WT GLUT1 or GLUT1 C207S in the presence of palmitoyl-CoA. The palmitoylation levels of GLUT1 were analyzed by APE assays. The top band indicates the palmitoylated GLUT1 (PEG-GLUT1). **c** Flag-tagged WT rDHHC9 or rDHHC9 C169S was reconstitutively expressed in GBM cells with the knockout of endogenous DHHC9. Immunoblotting was performed using indicated antibodies. **d** DHHC9-knockout U87 or T98G cells rescued with Flag-tagged WT rDHHC9 or rDHHC9 C169S was metabolically labeled with 50 µM of alkynyl PA for 4 h. Palmitoylation levels of GLUT1 were analyzed by click reaction and streptavidin bead pulldown, followed by immunoblotting. **e** APE assay was performed to analyze the GLUT1 palmitoylation in DHHC9-knockout U87 or T98G cells with reconstituted expression of Flag-tagged WT rDHHC9 or rDHHC9 C169S. The top band indicates the palmitoylated GLUT1 (PEG-GLUT1). **f** DHHC9-knockout U87 or T98G cells with or without reconstituted expression of Flag-tagged WT rDHHC9 or rDHHC9 C169S were incubated with 1 µg ml$^{-1}$ of CTB conjugated with Alexa Fluor 594 for 5 min at 37 °C. GLUT1 cellular localization was visualized by immunofluorescent staining using the antibody against GLUT1 and the PM was marked by Alexa Fluor 594-conjugated CTB. PM plasma membrane. Scale bar, 20 µm. **g** Levels of GLUT1 in the PM and ICM fractions were analyzed by immunoblotting in DHHC9-knockout U87 or T98G cells rescued with Flag-tagged WT rDHHC9 or rDHHC9 C169S. ATP1A1, calnexin, and GAPDH served as the marker of the PM, ICM, and cytosol fraction, respectively. PM plasma membrane, ICM intracellular membrane, WCL whole-cell lysate. Representative results were obtained from at least three independent experiments with similar results. Source data are provided as a Source Data file.

activation[21,22]. To further examine whether GLUT1 palmitoylation is affected by activation of these kinases. We pretreated U87 cells with LY294002 and Gö−6983 to inhibit the kinase activity of AKT and PKC, respectively. Alkynyl palmitic acid incorporation assay demonstrated that inhibition of AKT and PKC did not alter GLUT1 palmitoylation (Supplementary Fig. 5e), suggesting that

GLUT1 palmitoylation is independent of AKT and PKC activation in GBM cells.

**DHHC9-mediated GLUT1 S-palmitoylation promotes glycolysis, growth, and colony formation of GBM cells.** To determine the role of DHHC9-mediated GLUT1 S-palmitoylation in regulating

glucose uptake, we performed the glucose uptake assays by using a glucose structurally similar compound 2-deoxyglucose (2-DG) (Fig. 4a). DHHC9 KO or GLUT1 KO significantly suppressed glucose uptake in U87 and T98G cells (Fig. 4a). Notably, this suppression was ablated by reconstituted expression of WT rDHHC9 or rGLUT1, but not rDHHC9 C169S or rGLUT1 C207S, respectively (Fig. 4a). In line with this finding, we detected a decreased glycolytic rate (Fig. 4b), glucose consumption (Fig. 4c), and lactate production (Fig. 4d) in DHHC9-KO or GLUT1-KO U87 and T98G cells and these decreases were restored by reconstituted expression of WT rDHHC9 or rGLUT1, but not rDHHC9 C169S or rGLUT1 C207S, respectively (Fig. 4b–d). These results show that DHHC9-mediated GLUT1 S-palmitoylation promotes glycolysis in GBM cells. In contrast, endogenous DHHC9 depletion by adenovirus-mediated shRNAs did not significantly alter the glucose uptake and glycolytic rate in NHAs (Supplementary Fig. 6a–c), suggesting that NHAs may use mechanisms beyond DHHC9-mediated GLUT1 S-palmitoylation to support their glycolysis.

To further determine the functional consequences of DHHC9-mediated GLUT1 S-palmitoylation, we performed growth and colony-formation analyses of U87 and T98G cells. As expected, DHHC9 KO or GLUT1 KO inhibited proliferation (Fig. 4e) and colony formation (Fig. 4f, g) of U87 and T98G cells. Reconstituted expression of WT rDHHC9 or rGLUT1 restored cell proliferation (Fig. 4e) and colony formation (Fig. 4f, g). In contrast, reconstituted expression of rDHHC9 C169S or rGLUT1 C207S did not have these effects on cells (Fig. 4e–g). These results demonstrate that DHHC9-mediated GLUT1 S-palmitoylation promotes the growth and colony formation of GBM cells.

**DHHC9-mediated GLUT1 S-palmitoylation promotes GBM tumorigenesis.** To investigate the functions of DHHC9-mediated GLUT1 S-palmitoylation in GBM development, we intracranially injected athymic nude mice with luciferase-expressing U87 cells with or without knockout of endogenous DHHC9 or GLUT1 and reconstituted expression of their WT counterparts, rDHHC9 C169S or rGLUT1 C207S (Fig. 5a). Bioluminescent imaging demonstrated that injection of DHHC9-KO or GLUT1-KO U87 cells resulted in significant inhibition of brain tumor growth in mice (Fig. 5a), which was accompanied by a considerably prolonged survival time (Fig. 5b). These alterations were abrogated by reconstituted expression of WT rDHHC9 or rGLUT1, but not rDHHC9 C169S or rGLUT1 C207S (Fig. 5a, b).

Immunohistochemical (IHC) analyses with anti-DHHC9, anti-GLUT1, anti-Ki67, and anti-cleaved PARP1 antibodies revealed that tumor samples derived from U87 cells with the knockout of endogenous DHHC9 or GLUT1 displayed no expression of DHHC9 or GLUT1, respectively, had decreased expression of proliferation marker Ki67 and increased positive rate of apoptotic marker cleaved PARP1 (Fig. 5c). Reconstituted expression of WT rDHHC9 or rGLUT1, but not rDHHC9 C169S or rGLUT1 C207S, restored Ki67 expression, and abrogated the increased PARP1 cleavage (Fig. 5c). Notably, we found that knockout of endogenous DHHC9, similar to rGLUT1 C207S expression, significantly reduced S-palmitoylation (Supplementary Fig. 7a) and PM localization of GLUT1 (Fig. 5c). These effects were abrogated by reconstituted expression of WT rDHHC9, but not rDHHC9 C169S (Fig. 5c and Supplementary Fig. 7a). In addition, 2-DG uptake was markedly suppressed in tumors derived from U87 cells with the knockout of endogenous DHHC9 or GLUT1, and these suppression effects were abrogated by reconstituted expression of WT rDHHC9 or rGLUT1, but not rDHHC9 C169S or rGLUT1 C207S (Supplementary Fig. 7b). These results show that DHHC9-mediated GLUT1 S-palmitoylation promotes GBM tumorigenesis.

**DHHC9 expression positively correlates with GLUT1 plasma membrane localization in GBM specimens and indicates clinical aggressiveness of GBM.** To determine the clinical significance of the observed DHHC9-regulated GLUT1 PM localization, we next performed IHC analyses in GBM samples from 68 patients with anti-DHHC9 and anti-GLUT1 antibodies (Fig. 6a), revealing that DHHC9 expression levels were positively correlated with the percentage of GLUT1 PM localization (Fig. 6a). Quantification of the staining on a scale of 0–8 indicated that these correlations were significant (Fig. 6b). In addition, we performed the survival analyses in these patients, all of whom had received standard therapies, with stratification by levels of DHHC9 expression and PM-localized GLUT1 in tumor tissues (Fig. 6c). The median overall survival (OS) duration was 1057 and 1380 days for patients whose tumors had low levels of DHHC9 expression and PM-localized GLUT1, respectively, and 778 and 436 days for those whose tumors had high levels of DHHC9 expression and PM-localized GLUT1, respectively (Fig. 6c). Multivariate analyses revealed that a high level of DHHC9 expression and PM-localized GLUT1 was an independent, unfavorable prognostic indicator for OS of GBM patients after adjusting for patient age, sex, and total resection status, all of which are relevant clinical covariates (Supplementary Table 1). Taken together, these analyses reveal that a high level of DHHC9 expression and PM-localized GLUT1 significantly correlate with the clinical aggressiveness of GBM.

## Discussion
A previous study has shown that the P485L mutation creating a dileucine motif in the cytosolic tail causes GLUT1 internalization from the PM, leading to deficiency of glucose uptake in mammalian cells[23]. This phenomenon suggested that PM localization, which is independent of gene expression levels, is important for the biological functions of GLUT1. We demonstrate here that DHHC9 palmitoylates GLUT1 at Cys207 to maintain PM localization of GLUT1. Furthermore, PM-localized GLUT1 increases glucose uptake, glycolytic rate, and lactate production, consequently promoting GBM cell proliferation and GBM tumorigenesis (Fig. 6d). These findings expand the layers of protein biological functions regulated by post-translational modification, highlighting the importance of protein subcellular localization during cancer progression.

Protein S-palmitoylation was discovered as a post-translational modification that is well-known to regulate membrane distribution of proteins and cell signaling[10–12]. However, very little is understood about how S-palmitoylation contributes to metabolic homeostasis. This study demonstrates that DHHC9-mediated S-palmitoylation of glucose transporter GLUT1 promotes glycolysis in human GBM cells, implying that S-palmitoylation-regulated membrane localization of cellular proteins may be required for metabolic reprogramming in cancer cells.

Our findings demonstrate that knockout of palmitoyltransferase DHHC9 or disruption of DHHC9-mediated GLUT1 S-palmitoylation by site-specific point mutation inhibited growth and colony formation of GBM cells in vitro and GBM tumorigenesis in vivo. These results indicate that DHHC9 contributes to the malignant progression of GBM through facilitating GLUT1 palmitoylation.

GBM cells had a dramatic decrease in glucose uptake and glycolytic rate under the condition of DHHC9 KO. In contrast, NHAs, which have lower expression of DHHC9 compared to GBM cells, did not show significant alteration in glucose uptake or glycolytic rate when endogenous DHHC9 was depleted. These findings suggest that GBM cells have increased dependence

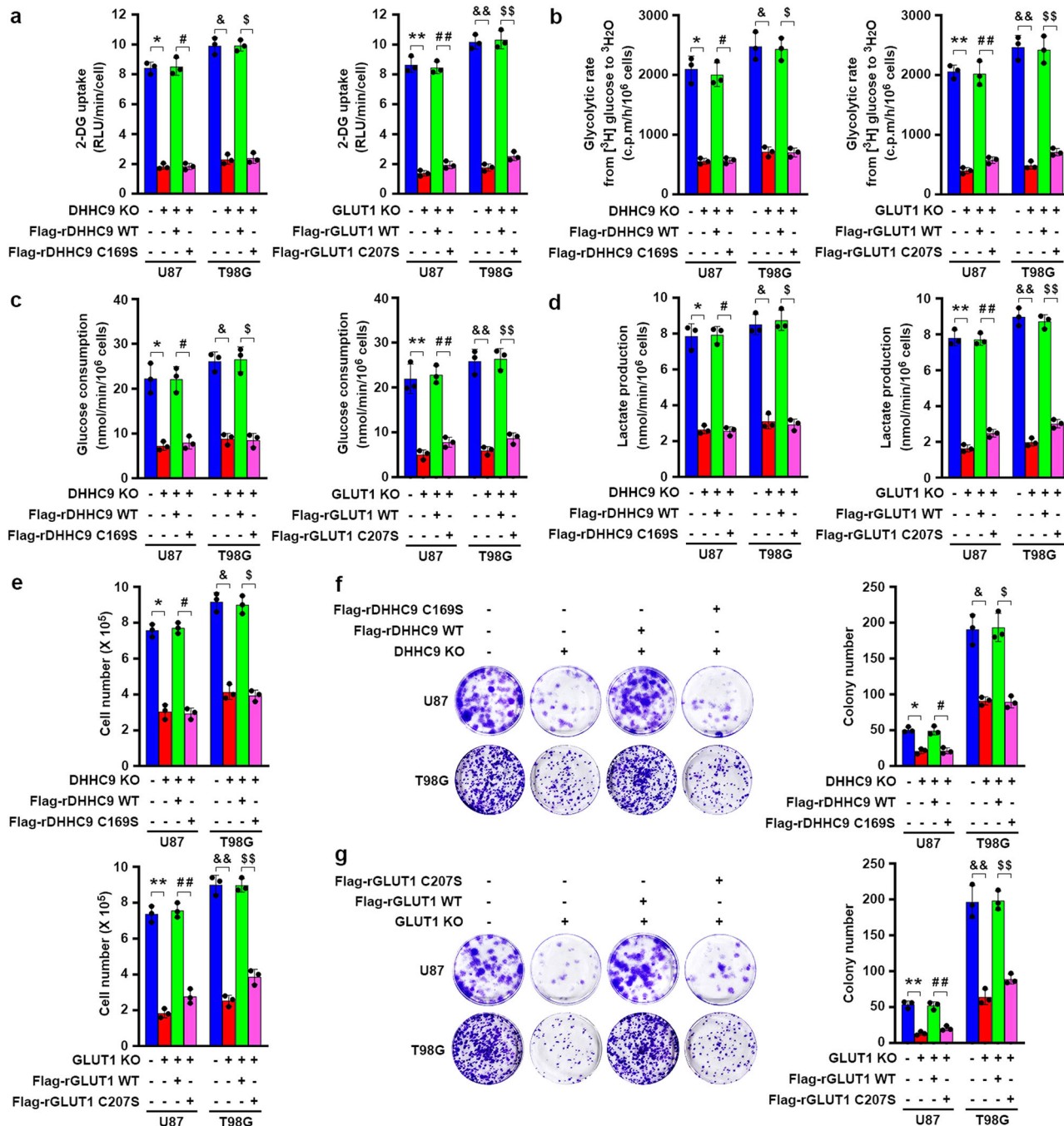

**Fig. 4 DHHC9-mediated GLUT1 S-palmitoylation promotes glycolysis, growth and colony formation of GBM cells.** Flag-tagged WT rDHHC9 or rDHHC9 C169S, WT rGLUT1, or rGLUT1 C207S were reconstitutively expressed in endogenous DHHC9-knockout or GLUT1-knockout U87 or T98G cells, respectively. Data represent the mean ± SD of three independent experiments. *P* value was determined by the two-tailed Student's *t* test. (**a**) The indicated cells were treated with 1 mM of 2-DG for 10 min. Uptake of 2-DG was measured using a glucose uptake kit and normalized to cell number. RLU relative luminescence units. *$P = 1.15E-05$, #$P = 4.81E-05$, &$P = 3.25E-05$, $$P = 1.28E-05$, **$P = 2.32E-05$, ##$P = 1.85E-05$, &&$P = 1.04E-05$, $$$P = 3.60E-05$. **b** The indicated cells were incubated with 5.5 mM glucose spiked with 10 μCi of D-[5-$^3$H] glucose for 1 h. The glycolytic rate was measured by monitoring the conversion of D-[5-$^3$H] glucose to $^3H_2O$ and normalized to cell numbers. c.p.m. counts per minute. *$P = 2.85E-04$, #$P = 2.54E-04$, &$P = 2.44E-04$, $$P = 1.21E-04$, **$P = 1.94E-05$, ##$P = 2.81E-04$, &&$P = 7.95E-05$, $$$P = 2.23E-04$. **c** The indicated cells were treated with DMEM without serum for 16 h. The media were collected for analysis of glucose consumption. *$P = 1.68E-03$, #$P = 1.40E-03$, &$P = 2.40E-04$, $$P = 5.93E-04$, **$P = 1.03E-03$, ##$P = 3.33E-04$, &&$P = 2.60E-04$, $$$P = 3.03E-04$. **d** The indicated cells were treated with DMEM without serum for 6 h. The media were collected for analysis of lactate production. *$P = 2.45E-04$, #$P = 5.98E-05$, &$P = 1.79E-04$, $$P = 9.70E-05$, **$P = 2.40E-05$, ##$P = 2.07E-05$, &&$P = 2.37E-05$, $$$P = 2.55E-05$. **e** The indicated cells ($2 \times 10^5$) were seeded in six-well plates and cultured for 3 days. The cells were then collected and counted. *$P = 1.26E-04$, #$P = 4.26E-05$, &$P = 1.91E-04$, $$P = 1.16E-04$, **$P = 4.96E-05$, ##$P = 1.30E-04$, &&$P = 5.21E-05$, $$$P = 9.62E-05$. **f**, **g** One thousand U87 or T98G cells were seeded in a 60-mm dish and cultured for 14 days. The cells were then fixed by 4% paraformaldehyde and stained with crystal violet (left panels). Colony-formation number in each dish was counted (right panels). **f** *$P = 6.01E-04$, #$P = 2.52E-03$, &$P = 1.27E-03$, $$P = 1.08E-03$. **g** **$P = 2.27E-04$, ##$P = 8.97E-04$, &&$P = 8.04E-04$, $$$P = 2.29E-04$. Source data are provided as a Source Data file.

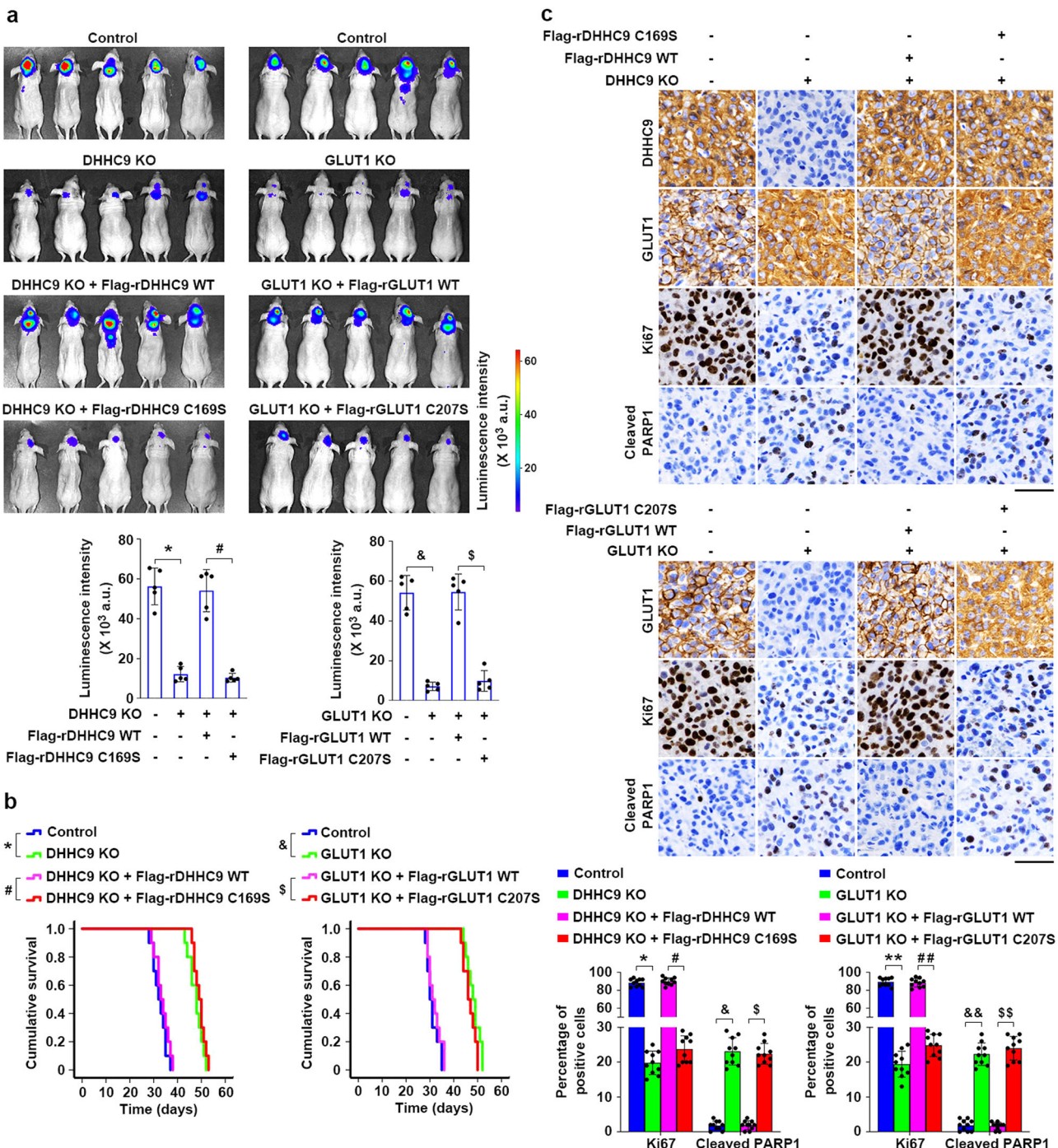

**Fig. 5 DHHC9-mediated GLUT1 S-palmitoylation promotes GBM tumorigenesis. a, b** Luciferase-expressing DHHC9-knockout U87 cells rescued with Flag-tagged WT rDHHC9 or rDHHC9 C169S and GLUT1-knockout GBM cells rescued with Flag-tagged WT rGLUT1 or rGLUT1 C207S were intracranially injected into athymic nude mice ($n = 5$). Luminescence intensity derived from tumors was measured and relative luminescence intensity is shown (**a**). Data were collected from $n = 5$ mice per group, and the mean ± SD values are shown (**a**). The survival times of the mice ($n = 10$) were recorded (**b**). **a** *$P = 2.57E{-}06$, #$P = 1.16E{-}05$, &$P = 9.63E{-}06$, $$$P = 1.82E{-}05$. **b** *$P = 4.00E{-}06$, #$P = 3.00E{-}06$, &$P = 5.00E{-}06$, $$$P = 4.00E{-}06$. **c** Representative images of immunohistochemical staining of DHHC9, GLUT1, Ki67, and cleaved PARP in paraffin-embedded xenograft tumor tissues collected from the indicated groups (top and middle). Scale bar, 50 μm. The expression levels of Ki67 and cleaved PARP1 were quantified for ten microscopic fields of the tumor samples (bottom). The mean ± SD values are shown. *$P = 3.87E{-}19$, #$P = 1.44E{-}18$, &$P = 3.33E{-}12$, $$$P = 8.58E{-}14$, **$P = 1.15E{-}18$, ##$P = 4.84E{-}18$, &&$P = 5.63E{-}13$, $$$$P = 1.14E{-}13$. $P$ values were determined by the two-tailed Student's $t$ test (**a**, **c**) and the two-tailed log-rank test (**b**). a.u., arbitrary unit (**a**). Source data are provided as a Source Data file.

on DHHC9-mediated GLUT1 S-palmitoylation, unlike normal astrocytes which primarily depend on other mechanisms to maintain their glycolysis, highlighting that DHHC9 could be a potential target for cancer cell-specific inhibition of glycolysis.

Therefore, the development of small molecules against DHHC9 palmitoyltransferase activity may provide an alternative approach to treat human GBM. Importantly, high DHHC9 expression in tumor tissues appeared to correlate with poor survival of GBM

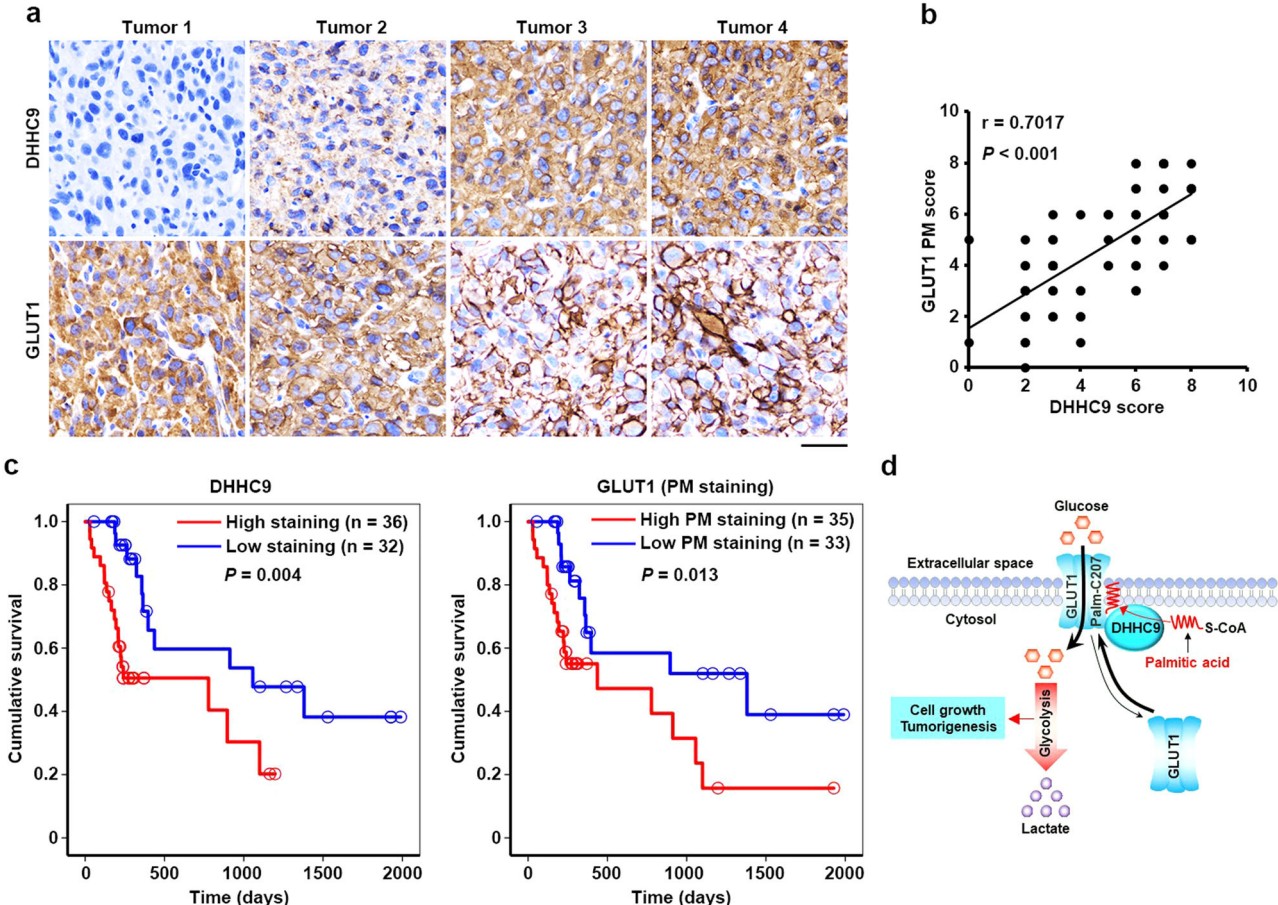

**Fig. 6 DHHC9 expression positively correlates with GLUT1 PM localization in GBM specimens and indicates clinical aggressiveness of GBM. a**, **b** Sixty-eight human primary GBM specimens were immunohistochemically stained with indicated antibodies. Representative photos of tumors are shown (**a**). Immunohistochemistry staining scores of DHHC9 and PM-localized GLUT1 were analyzed by the two-tailed Pearson correlation (**b**). Note that some of the dots on the graphs represent more than one specimen (i.e., some scores overlapped). **c** Kaplan–Meier method was used to plot survival curves in human GBM specimens ($n = 68$) with high (5–8 staining scores, red curve) and low (0–4 staining scores, blue curve) levels of DHHC9 and PM-localized GLUT1. The two-tailed log-rank test was used to compare the overall survival rate. Empty circles represent censored data from patients alive at the last clinical follow-up. **d** A mechanism of S-palmitoylated GLUT1-dependent glycolysis. GLUT1 Cys207 palmitoylation catalyzed by DHHC9 resulted in PM localization of GLUT1, leading to enhanced uptake of glucose, thereby promoting glycolysis, cell growth, and GBM tumorigenesis. $P$ values were calculated by the two-tailed Pearson correlation (**b**) and the two-tailed log-rank test (**c**). Scale bar, 20 μm (**a**). Source data are provided as a Source Data file.

patients, suggesting that DHHC9 could serve as a prognostic indicator for GBM patients.

## Methods

**Materials.** Rabbit monoclonal antibodies recognizing GLUT1 (#ab115730), tubulin (#ab176560), Ki67 (#ab92742) and ATP1A1 (#ab76020), and mouse monoclonal antibody recognizing GAPDH (#ab8245) were obtained from Abcam. Mouse monoclonal antibodies recognizing GCP16 (#sc-101278) and GLUT3 (#sc-74399), n-dodecyl-β-D-maltoside (DDM) (#sc-281071), and n-Nonyl-β-D-gluco-pyranoside (#sc-281084) were purchased from Santa Cruz Biotechnology. Mouse monoclonal antibody against Flag tag (#F3165), and rabbit polyclonal antibodies recognizing GLUT1 pS226 (#ABN991), nickel affinity gel (#P6611), anti-Flag M2 affinity gel (#A2220), 2-deoxy-D-glucose (2-DG) (#D8375), palmitic acid (#P0500), dithiothreitol (#D9779), ethylenediaminetetraacetic acid (EDTA) (#E6758), phe-nylmethanesulfonyl fluoride (PMSF) (#P7626), leupeptin (#L2884), 2-Bromohexadecanoic acid (2-BP) (#21604), hydroxylamine (#159417), palmos-tatin B (#178501), N-Ethylmaleimide (NEM) (#04260), mPEG-maleimide 5 kDa (#63187), TCEP (#646547), crystal violet (#C0775), and IPTG were purchased from Sigma. Mouse monoclonal antibody recognizing GST (#2624), and rabbit mono-clonal antibodies against hemagglutinin (HA) tag (#3724), calnexin (#2679), cleaved PARP (#5625), AKT pT308 (#13038), and AKT (#4691) were purchased from Cell Signaling Technology. Rabbit polyclonal antibodies recognizing DHHC9 (#PA5-56868), 10 kDa spin column (#88513), streptavidin agarose beads (#SA10004), EDTA-free protease inhibitors cocktail (#A32965), cholera toxin subunit B (CTB) conjugated with Alexa Fluor 594 (#C34777), horseradish peroxidase-conjugated goat anti-mouse (#G-21040) and anti-rabbit (#G-21234)

secondary antibodies, Alexa Fluor 488 (#A-11008)/594 (#A-11012)-conjugated goat anti-rabbit secondary antibodies, glutathione agarose, and 4',6-diamidino-2-phenylindole (DAPI) were obtained from Thermo Fisher Scientific. Lipofectamine 3000 transfection reagents were obtained from Invitrogen. Palmitoyl alkyne-CoA (#15968) and alkynyl arachidonic acid (#10538) were obtained from Cayman. Alkynyl palmitic acid (#1165), alkynyl myristic acid (#1164), alkynyl stearic acid (#1166), and biotin picolyl azide (#1167) were purchased from Click Chemistry Tools. D-[5-³H]-glucose was purchased from PerkinElmer.

**Cell lines and cell culture conditions.** Human GBM cell lines including U87, T98G, LN229, LN18, A172, and luciferase-expressing U87 cells were maintained in Dulbecco's modified Eagle's medium (DMEM) supplemented with 10% bovine calf serum (HyClone). Normal Human Astrocytes (NHAs) (#CC-2565) obtained from Lonza were cultured in Astrocyte Growth Medium containing basal medium and necessary growth factors. Cell lines used in this study were not found in the BioSample database of commonly misidentified cell lines provided by the Inter-national Cell Line Authentication Committee (ICLAC). Experiments involving glycolytic flux were performed with a maintenance medium supplemented with physiologic concentrations of glucose (5.5 mM). The medium was changed daily. Cell lines were authenticated by Short Tandem Repeat (STR) profiling and were routinely tested for mycoplasma contamination.

PDX cells derived from a GBM primary tissue were obtained from Drs. Chunsheng Kang and Chuan Fang (Tianjin Medical University) and used in previous publications[24,25]. The GBM tissue was surgically removed from a 46-year-old male Chinese patient. Informed consent was obtained and the use of a patient specimen and relevant database was approved by the Human Research Ethics Committee of Tianjin Medical University.

**Glycolytic rate assays**. Glycolytic rate assay was performed according to the published procedure with slight modifications[26]. Briefly, GBM cells ($5 \times 10^5$) see-ded in a 6-well plate were washed once with PBS and incubated in 2 ml of Krebs buffer (126 mM NaCl, 2.5 mM KCl, 25 mM NaHCO$_3$, 1.2 mM NaH$_2$PO$_4$, 1.2 mM MgCl$_2$, 2.5 mM CaCl$_2$) without glucose for 30 min at 37 °C. The Krebs buffer was removed and 2 ml of Krebs buffer containing 5.5 mM glucose spiked with 10 µCi of D-[5-$^3$H]-glucose was then added into each well followed by incubation for 1 h at 37 °C. An aliquot of Krebs buffer (50 µl) was mixed with an equal volume of 0.2 N HCl in an uncapped PCR tube, which was then transferred into a 1.5-ml Eppendorf tube containing 0.5 ml of unlabeled distill water. The Eppendorf tubes were sealed to allowed diffusion of $^3$H$_2$O into unlabeled water for 24 h at 37 °C. The amount of diffused $^3$H$_2$O was measured using a scintillation counter and normalized to the cell number counted in each sample.

**Glucose uptake assay**. Glucose uptake was measured by a kit obtained from Promega (#J1342) according to the manufacturer's instructions. Briefly, 10,000 GBM cells were seeded into 96-well plates and grew for 12 h. The cells were washed twice with PBS and incubated with 1 mM of 2-DG for 10 min at 37 °C. The uptake was terminated by the addition of an acid detergent solution (stop buffer) and neutralization buffer was then added to neutralize the acid. The 2-DG6P detection reagent containing glucose-6-phosphate dehydrogenase, NADP$^+$, reductase, recombinant luciferase, and proluciferin substrate was added to the sample wells. The plate was incubated for 1 h at 25 °C and luminescence intensity was read on a luminometer with 0.3–1 s integration. The glucose uptake level was normalized according to the cell number counted in a duplicate sample.

**Production and purification of recombinant proteins**. Baculovirus expression system (Thermo Fisher Scientific, MA) was used for the production of 6×His-tagged GLUT1 and GST-tagged DHHC9 or GCP16 proteins. The cells were har-vested 48 h after viral infection and homogenized in the buffer (25 mM Tris [pH 8.0], 150 mM NaCl) containing protease inhibitor cocktail and 2% (w/v) n-dodecyl-β-D-maltoside (DDM) at 4 °C. Cell debris was removed by low-speed centrifugation for 10 min. For purification of His-tagged proteins, the supernatant was loaded onto a Ni-NTA column (GE Healthcare Life Sciences, PA) followed by washing with five column volumes of 20 mM imidazole and subsequent elution with 250 mM imidazole in the presence of 0.05% (w/v) DDM and 5% glycerol (w/v). For purification of GST-tagged proteins, the supernatant was loaded onto a GSTrap HP column (GE Healthcare Life Sciences) followed by washing with five column volumes of phosphate-buffered saline (PBS) and subsequent elution with 10 mM reduced glutathione. The partially purified proteins in 10 kDa spin columns were desalted by washing twice with ice-cold buffer (25 mM MES [pH 6.0], 150 mM NaCl, 5% glycerol, 0.4% (w/v) n-Nonyl-β-D-glucopyranoside) and con-centrated to about 10 mg ml$^{-1}$ followed by loading onto a Superdex-200 gel fil-tration column (GE Healthcare Life Sciences, PA) to remove the contaminated proteins. The peak fractions were collected and separated by SDS-PAGE. High-sensitivity colloidal *Coomassie* blue (G-250) staining was performed to examine the purification efficiency.

**Transient transfection**. The plasmids were transiently transfected into GBM cells with Lipofectamine 3000 (Invitrogen, CA) according to the manufacturer's pro-tocol. Cells were harvested and analyzed 36 h after transfection.

**Immunoprecipitation and immunoblotting**. Extraction of proteins from cultured cells was performed with a modified lysis buffer (50 mM Tris-HCl [pH 7.5], 0.1% SDS, 1% Triton X-100, 150 mM NaCl, 1 mM dithiothreitol, 0.5 mM EDTA, 100 µM PMSF, 100 µM leupeptin). Cell debris was removed by centrifugation at 13,400×*g* for 10 min at 4 °C. The supernatants (2 mg protein/ml) were incubated with the specific antibodies overnight and then mixed with protein A or G agarose beads for 1 h at 4 °C. Immunocomplexes washed with lysis buffer three times were subjected to immunoblot analyses with corresponding antibodies as described previously[27].

For immunoblotting, samples were resolved on 8%, 10%, or 12% polyacrylamide minigels (Bio-Rad) and transferred onto a PVDF membrane (GE Healthcare Life Sciences, PA) by wet or semi-dry transfer. The membranes were probed with primary and then HRP-conjugated secondary antibodies. Immunoblots were visualized using SuperSignal West Pico Chemiluminescent Substrate (Thermo Fisher Scientific, MA) and the band intensity was quantified by the Image Lab software program (Bio-Rad Laboratories, CA).

**Metabolic labeling, click reaction, and streptavidin pulldown**. Metabolic labeling, click reaction, and streptavidin pulldown were performed according to the published procedure with slight modifications[28]. Briefly, cells labeled with alkynyl probes were washed with ice-cold D-PBS [2.67 mM KCl, 138 mM NaCl, 1.47 mM KH$_2$PO$_4$, 8.1 mM Na$_2$HPO$_4$, pH 7.4] and lysed in lysis buffer [50 mM Tris-HCl [pH 7.4], 150 mM NaCl, 0.1% Triton X-100, 0.1% SDS] supplemented with pro-tease inhibitor cocktail, 1 µM palmostatin B and 50 mM NEM. The cell lysate was clarified at 13,400×*g* for 10 min at 4 °C and the supernatant was subjected to a Cu(I)-assisted click reaction with biotin picolyl azide followed by loading onto a 10 kDa spin column to remove the free biotin picolyl azide using lysis buffer. Streptavidin agarose beads were incubated with the samples for 2 h at room

temperature and then washed three times with lysis buffer. The bound proteins were eluted by boiling with SDS-PAGE sample buffer without DTT for 10 min at 95 °C, and then analyzed by immunoblotting.

**In vitro protein acyltransferase (PAT) assay**. Palmitoyl acyltransferase con-stituted by purified recombinant DHHC9 and GCP16 proteins (100 ng for each) were incubated with GLUT1 (2 µg) in 25 µl of reaction buffer (50 mM Tris-HCl [pH 7.4], 10 µM palmitoyl alkyne-CoA, 1 µM palmostatin B) at 25 °C for 1 h followed by a Cu(I)-assisted click reaction with biotin picolyl azide (50 µM) to biotinylate the proteins with the incorporation of palmitoyl alkyne. The samples were load onto 10 kDa spin columns to remove the free biotin picolyl azide using a buffer containing 50 mM Tris-HCl [pH 7.4], 150 mM NaCl, 0.5 mM EDTA, 0.1% Triton X-100, 0.1% SDS, and 0.5% NP40. Biotinylated proteins were captured by streptavidin agarose beads prior to boiling in SDS-PAGE sample buffer without DTT for 10 min at 95 °C. Immunoblotting was performed to analyze the palmi-toylation of the target protein.

**GST pull-down assay**. His-tagged purified proteins (200 ng/sample) were mixed with 100 ng of GST fusion proteins or GST protein as a control in a binding buffer (50 mM Tris-HCl [pH 7.5], 1% Triton X-100, 150 mM NaCl, 1 mM DTT, 0.5 mM EDTA, 100 µM PMSF, 100 µM leupeptin) for 1 h and incubated with glutathione agarose beads for an additional 30 min at 4 °C. The bound protein complexes were retained by washing the glutathione beads with binding buffer three times and detected by immunoblots.

**Confocal microscopic analysis**. To label the plasma membrane of cells, GBM cells with or without expression of eGFP-tagged proteins were seeded into eight-well chamber slides 12 h before staining. Cholera toxin subunit B (1 µg ml$^{-1}$) conjugated with Alexa Fluor 594 (Thermo Fisher Scientific, MA) was incubated with the cells in DMEM supplemented with 10% bovine calf serum for 5 min at 37 °C. Unbound Alexa Fluor 594-CTB was washed away by cold PBS, and cells were fixed with 4% paraf-ormaldehyde for 20 min at room temperature. For immunofluorescent staining, fixed cells were incubated with primary antibodies at a dilution of 1:100 overnight at 4 °C, followed by staining with secondary antibodies conjugated with Alexa Fluor 488 for 30 min. The cells were then counterstained with 1 µg ml$^{-1}$ of DAPI and mounted using ProLong Gold antifade reagent (Thermo Fisher Scientific). Images of the cells were acquired using an LSM700 inverted microscope equipped with a ×40 objective (Zeiss, NY). Photoshop CS6 (Adobe Systems, CA) software program was used for manual quantitation of the images.

**Plasma membrane and intracellular membrane fractionation**. Plasma mem-brane (PM) and intracellular membrane (ICM) fractions were isolated using a PM/ICM protein extraction kit (#ab65400) obtained from Abcam according to the manufacturer's instructions. Briefly, cells ($5 \times 10^8$) cultured on 15-cm dishes were rinsed twice with ice-cold PBS and immediately incubated with 2 ml of homo-genized buffer on ice and harvested using a cell scraper. Resuspended cells were homogenized in a Dounce homogenizer for 30–50 times on ice and spun at 700×*g* for 5 min at 4 °C. The supernatants were transferred to new vials and centrifuged at 10,000×*g* for 30 min at 4 °C. The pellet and supernatant contained total cellular membrane proteins and cytosolic proteins, respectively. The pellet was resuspended in 200 µl of upper-phase solution and then mixed with 200 µl of lower-phase solution. The samples were incubated on ice for 5 min and centrifuged at 1000×*g* for 5 min at 4 °C; upper-phase solution and lower-phase solution containing PM proteins and ICM proteins, respectively, were collected and transferred to new tubes. Extractions were repeated by adding fresh upper-phase solution to the lower-phase solution and vice versa. The resulting upper-phase solution and the lower-phase solution from the repeated extraction were combined and diluted with 5×volume of water, followed by centrifuge at 15,000×*g* for 30 min at 4 °C. The pellets containing PM or ICM proteins were harvested and then analyzed by immunoblotting.

**Lentiviral production and generation of stable cell lines**. Guide RNAs targeting *SLC2A1* (GLUT1 coding gene) or *DHHCs* were designed using Cas9 target design online tool (http://www.genome-engineering.org). Three different promising guide RNAs were chosen and cloned into the lentiCRISPRv2 lentiviral vector with a selectable marker of puromycin. The empty vector lentiCRISPRv2 was used as a control. The presence of the guide RNA was confirmed by sequencing and lenti-viruses were produced by cotransfecting 293FT cells with lentiCRISPRv2 plasmid containing expression cassettes of hSpCas9 and the chimeric guide RNA, and two packaging plasmids pMD2.G (#12259) and psPAX2 (Addgene #12260). Infectious lentiviruses were harvested 72 h after transfection, centrifuged to remove cell debris, and filtered through a 0.45-µm filter (Millipore, MA). GBM cells were infected with lentiviruses at a multiplicity of infection [MOI] of 1 and then selected by 1 µg ml$^{-1}$ of puromycin for 7 days. The knockout efficiency was evaluated by immunoblotting. For rescued expression of GLUT1 or DHHC9 in GLUT1 and DHHC9-knockout cells, respectively, Flag-tagged sgRNA-resistant (r) wild-type GLUT1 or rDHHC9 or their mutants (rGLUT1 C207S and rDHHC9 C169S) were cloned into pCDH lentiviral vector selectable marker of hygromycin. Lentiviruses were produced by the pMD2.G/psPAX2 packaging system. Infection of cells was

performed in 6-well plates at a MOI of 1, and selected under 200 µg ml$^{-1}$ hygromycin for 10 days. The rescued protein expression was evaluated by immunoblotting using specific antibodies.

The pGIPZ lentiviral vector was used for the construction of the control shRNA and shRNAs targeting *DHHC9*. The lentiviruses were produced as described above. GBM cells were infected with the lentiviruses followed by selection with 1 µg ml$^{-1}$ of puromycin for 7 days. The knockdown efficiency was evaluated by immunoblotting using specific antibodies.

Supplementary Table 2 contains the detailed information of the nucleotide sequences of guide RNAs (gRNAs) or shRNAs targeting indicated genes.

**Adenovirus preparation and infection.** Recombinant adenoviruses expressing the control shRNA or *DHHC9* shRNAs were produced by RAPAd adenoviral expression system (Cell Biolabs, CA) according to the manufacturer's instructions. Briefly, an XhoI/EcoRI fragment containing a U6-shRNA cassette was amplified from pLKO.1 vector using high-fidelity PCR and ligated into XhoI/EcoRI-digested adenoviral shuttle vector pacAd5 K-NpA. Adenoviruses were produced by cotransfection of the PacI-digested pacAd5 U6-shRNA and pacAd5 9.2-100 plasmid into Adeno-X 293 cells (Clontech Laboratories, CA). Infectious adenoviruses were collected 8 days after transfection, centrifuged to remove cell debris, filtered through a 0.45-µm filter (Millipore), and used to infect the Adeno-X 293 cells to produce high-titer adenoviral particles. Adenoviruses were purified using an Adenovirus Purification Kit (Clontech Laboratories) and titration was determined by an Adeno-X Rapid Titer Kit (Clontech Laboratories) according to the manufacturer's instructions.

For adenovirus infection, PDX cells or NHAs were infected with indicated adenoviruses at a multiplicity of infection (MOI) of 30 and replaced with fresh DMEM 6 h after viral infection.

**Acyl-PEG exchange (APE) assay.** APE assay was performed according to the published procedure with slight modifications[29]. In brief, trypsinized cells were washed three times with PBS and lysed in lysis buffer (5 mM triethanolamine, 150 mM NaCl, 4% SDS, 2 units benzonase, 5 mM PMSF, pH 7.4) with protease inhibitor cocktail followed by adding EDTA to a final concentration of 5 mM. The cell lysates were treated with 10 mM of TCEP for 30 min and then incubated with 25 mM of NEM for 2 h at room temperature to reduce disulfide bonds and cap the free cysteine residues, respectively. The mixture was precipitated by sequential addition of methanol, chloroform, and distilled water (4:1.5:3, relative to sample volume) into the 1.5-ml Eppendorf tube. The protein pellets were washed twice with prechilled methanol, resuspended in TEA buffer (5 mM Triethanolamine, 150 mM NaCl, 4% SDS, 4 mM EDTA, pH 7.4), and then incubated with 0.75 M of NH$_2$OH for 1 h at room temperature to cleave palmitoylation thioester bonds. NH$_2$OH was removed by methanol-chloroform-water precipitation and the protein pellets resuspended in TEA buffer with 0.2% Triton X-100 were incubated with 1 mM of mPEG-Mal (5 kDa) for 2 h at room temperature. The samples were precipitated again by methanol-chloroform-water and then resuspended with 1× Laemmli buffer without DTT, boiled at 95 °C for 3 min, separated by SDS-PAGE, and analyzed by immunoblotting.

**DNA constructs and mutagenesis.** PCR-amplified human *SLC2A1*, *DHHC9*, and *GCP16* were subcloned into baculoviral vector pFastBac1, lentiviral vector pCDH (hygromycin resistance), bicistronic expression vector pIRES, or green fluorescent protein vector pEGFP-C1. pFastBac1 6×His-GLUT1 C207S, GST-DHHC9 C169S and pCDH Flag-GLUT1 C207S, Flag-DHHC9 C169S, and pEGFP-GLUT1 C207S were constructed using a QuikChange site-directed mutagenesis kit (Stratagene). To eliminate the protospacer-adjacent motif (PAM) sequence of NGG, which is required for *Streptococcus pyogenes* Cas9 (SpyCas9)-mediated DNA cleavage, amino acid codon of GLUT1 E220 and DHHC9 S238 were converted from GAG to GAA (G877A) and TCC to AGT (T1106A/C1107G/C1108T) to generate expressing plasmids pCDH Flag-rGLUT1 and pCDH Flag-rDHHC9, respectively.

**Colony-formation assay.** One thousand cells were seeded in a 60-mm plate and cultured DMEM supplemented with 10% bovine calf serum for 2 weeks. The cells were then fixed with 4% formaldehyde, stained with 2% crystal violet diluted in water, and photographed.

**Cell proliferation assay.** A total of $2 \times 10^5$ cells suspended in 2 ml of medium were plated and maintained in DMEM supplemented with 10% bovine calf serum. The cells in each well were resuspended by trypsinization and counted every day after seeding.

**Measurements of glucose consumption and lactate production.** Glucose consumption and lactate production were measured as the published procedure with slight modifications[30]. Briefly, attached GBM cells ($1 \times 10^6$) seeded in 60-mm dishes were changed to 3 ml of serum-free DMEM, 6 and 16 h later, the culture medium was collected for measurements of lactate production and glucose consumption, respectively. The levels of lactate and glucose in the culture medium were determined by kits obtained from Promega (#J5022) and Sigma (#GAGO20), respectively, according to the manufacturer's instructions. Glucose consumption was calculated from the difference in glucose concentration in the medium with or

without cell incubation. Lactate production and glucose consumption were normalized to cell numbers.

**Xenograft study.** GBM cells ($5 \times 10^5$) suspended in 5 µl of DMEM were intracranially injected into female 4-week-old athymic nude mice (five mice/group) obtained from GemPharmatech (Nanjing, China). Animals injected with U87 cells were sacrificed 28 days after glioma cell injection. The brain of each mouse was harvested, fixed in 4% formaldehyde, and embedded in paraffin. Tumor formation and phenotype were determined by histologic analysis of H&E-stained sections. In another group of mice (ten mice/group), the survival time of each mouse was measured from the time point when intracranial injection of tumor cells was performed until the clear signs of morbidity appeared.

Glucose uptake assay in tumor tissue was performed at 28 days after intracranial injection of tumor cells. Briefly, 2-DG (0.2 ml [500 mg/kg]) was delivered to mice via intraperitoneal administration 6 h before mouse scarification. The brain of each mouse was then harvested, and 20 mg of tumor tissue was dissected for homogenization in 0.2 ml of lysis buffer, which is prepared by mixing two volumes of PBS with one volume of Stop buffer provided by the glucose uptake assay kit (Promega #J1342). Homogenate was cleared by centrifugation at 10,000×g for 10 min. The supernatant of each sample (75 µl) was transferred to a 96-well assay plate and mixed with 25 µl of neutralization buffer followed by the addition of 100 µl of 2-DG6P detection reagent. The plate was incubated for 1 h at 25 °C, and luminescence intensity was read by a luminometer with 0.3–1 s integration.

To detect the GLUT1 palmitoylation in xenograft tumor tissues, 100 mg of tumor tissue was dissected for homogenization in 0.5 ml of lysis buffer (5 mM triethanolamine, 150 mM NaCl, 4% SDS, 2 units benzonase, 5 mM PMSF, pH 7.4) with protease inhibitor cocktail. Cell debris was removed by centrifugation at 10,000×g for 10 min. The supernatant was transferred to a new tube for the APE assay.

Mice were housed in a pathogen-free environment with the temperature maintained at 23 ± 2 °C and relative humidity at 50 to 65% under a 12 h/12 h light/dark cycle with free access to food and water. The animals were housed at 3–5 mice per cage and treated in accordance with the Guide for the Care and Use of Laboratory Animals published by the National Academy of Sciences and the National Institutes of Health. The use of animals in this study was approved by the Institutional Animal Care and Use Committee of the Center for Animal Experiments of the Institute of Biophysics, Chinese Academy of Sciences.

**Bioluminescent imaging.** Bioluminescent imaging of mice was performed according to the published procedure with slight modifications[25]. Briefly, D-luciferin (450 mg/kg; Cayman Chemical, MI) in 250 µl of PBS was subcutaneously injected into the neck region of mice. Images of the mice were acquired 10–20 min after D-luciferin administration, the peak photon flux within a region of interest was recorded and quantified using an IVIS Lumina System coupled with the Living Image data-acquisition software program (Xenogen Corporation, Alameda, CA).

**Immunohistochemical staining.** Mouse tumor samples or GBM specimens collected with informed consent were fixed and prepared for IHC staining. The IHC staining was performed using a VECTASTAIN ABC kit (Vector Laboratories, CA) according to the manufacturer's instructions.

The tissue sections from paraffin-embedded human GBM specimens were stained with antibodies against GLUT1, DHHC9, or nonspecific IgG as a negative control. Xenograft tissues of mice were stained with antibodies against GLUT1, DHHC9, Ki67, cleaved PARP, or nonspecific IgG as a negative control. For DHHC9 staining, the staining score of the tissue sections was determined by the percentage of positive cells and staining intensity as previously defined[27], proportion scores were assigned by the following criteria: 0 if no tumor cells showed positive staining; 1 if 0–1%; 2 if 2–10%; 3 if 11–30%; 4 if 31–70%; and 5 if 71–100%, and the intensity scores of the staining were assigned by the following criteria: 0, negative; 1, weak; 2, moderate; and 3, strong. A total score (range 0–8) was obtained by combining the proportion and intensity scores. For PM GLUT1 staining, the percentage of PM GLUT1 staining was quantified for ten microscopic fields of the tumor tissue in each section, and the scores were assigned by the following criteria: 0 if no PM staining, 1 if 0–10%; 2 if 11–20%; 3 if 21–30%; 4 if 31–40%; 5 if 41–50%; 6 if 51–60%; 7 if 61–70%; and 8 if 71–100%. The tissue sections with staining scores of 0–4 and 5–8 were defined as low staining and high staining, respectively. Overall survival was defined as the time from date of diagnosis to death or last known date of follow-up. All patients had received standard clinical treatments. The use of patient specimens and the relevant database was approved by the Human Research Ethics Committee of the First Affiliated Hospital of Nanjing Medical University.

**Statistical analysis.** Statistical analyses were performed using IBM SPSS Statistics software. The two-tailed unpaired or paired Student's *t* test was used to analyze means ± SD between the control and experimental groups. The survival rate analyses were performed using the two-tailed log-rank test. The correlation between DHHC9 and PM-localized GLUT1 levels was determined by the two-tailed Pearson correlation coefficient. Multivariate analysis of overall survival in GBM patients was performed using Cox's regression model. *P* values less than 0.05 were considered significant.

**Reporting summary**. Further information on research design is available in the Nature Research Reporting Summary linked to this article.

## Data availability
Source data are provided with this paper.

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

## Acknowledgements
We thank Hongjie Zhang of the Core Facility of Protein Sciences, Institute of Biophysics for technical assistance. This work was supported by Key Program of the Chinese Academy of Sciences (Grant No. KJZD-SW-L05 to Xinjian Li); the National Natural Science Foundation of China (Grant No. 82073060 to Xinjian Li); the National Key R&D Program of China (Grant No. 2020YFC2002700 to Xinjian Li); the Thousand Young Talents Program; and the National Science Foundation for Young Scientists of China (Grant No. 82003032 to Z.Z.).

## Author contributions
Xinjian Li conceived and designed the study; Xinjian Li, Z.Z., Xin Li, F.Y., C.C., P.L., Y.R., P.S. and Z.W. performed the experiments; Y.Y. provided reagents and pathology assistance; Y.-X.Z. provided conceptual advice; Xinjian Li wrote the manuscript with comments from all authors.

## Competing interests
The authors declare no competing interests.
