## [Peer Review File · Nature Communications]

DHHC9-mediated GLUT1 S-palmitoylation promotes glioblastoma glycolysis and tumorigenesisREVIEWER COMMENTS

Reviewer #1 (Remarks to the Author); expert on glioblastoma and metabolism:

This manuscript investigates how palmitoylation affects GLUT1 function and tumorigenesis in GBM. The study demonstrates that 90% of Glut1 is palmitoylated at C207, which promotes its localization at the plasma membrane. DHHC9 was identified as the palmitoyl acyltransferase responsible for Glut1 palmitoylation. Also, DHHC9 promotes growth, glycolysis and colony formation of GBM cells by palmitoylation of Glut1. A catalytically inactive mutant of DHHC9 or Glut1 C207S abrogates Glut1 palmitoylation of Glut1 and its PM distribution resulting in reduced growth, glycolysis and colony formation. Similar results were obtained in an orthotopic in vivo U87 GBM model. A positive correlation between DHHC9 expression and Glut1 PM localization was identified and is associated with poor prognosis in patient samples. Based on these results, the authors conclude that DHHC9 contributes to GBM tumorigenesis and DHHC9 as a promising target of GBM therapies. While there has been a prior report that GLUT1 palmitoylation might be important in brain capillaries, this is the first study to show such regulation in GBM. It further is the first study to describe the specific site of palmitoylation and the enzyme responsible for placing the palmitoyl group on GLUT1. Because of the novelty and potential therapeutic implications of this finding, my enthusiasm for this work is high. However, there are some issues detailed below that should be addressed.

Major issues:

1. The rigorous molecular biology studies are outstanding, and all phenotypes seem identical in the two immortalized GBM cell lines (U87, T98G) used throughout this work. However, immortalized GBM cell lines do not recapitulate much of the known biology of GBM human tumors and should not be the only models used if the authors wish to claim "DHHC9-mediated palmitoylation of GLUT1 is important in GBM". I would ask that the authors validate their cell line findings in a patient-derived model of GBM. Options for such a study could include patient-derived neurospheres, patient-derived adherent cells that have not been in culture for decades (<https://www.ncbi.nlm.nih.gov/pmc/articles/PMC4634360/>), or patient-derived xenografts. Due to the strength of rest of the paper and inclusion of robust human tumor data, I do not feel that additional in vivo animal work is needed, just a validation that deletion/knockdown of DHHC9 reduces GLUT1 PM localization in a better model than U87 or T98G.
2. I am not sure whether the regulation the authors describe is GBM-specific or also happens in normal brain. Because GLUT1 function is very important for normal brain (<https://academic.oup.com/brain/article/133/3/655/275024>), this distinction will be important for whether this regulation is a useful therapeutic strategy (i.e., it is GBM specific), or if it is simply a very interesting mechanistic finding. Studies regarding DHHC9-mediated GLUT1 palmitoylation in a non-transformed normal human astrocyte model could help address this issue.

Minor issues:

1. I would suggest a slight re-wording of the title to: DHHC9-mediated GLUT1 S-palmitoylation promotes glioblastoma glycolysis and tumorigenesis
2. Fig 4 and 5: Can the authors please add statistical significance (*) and the corresponding statistical tests with p-values in all the figure legends in Fig 4 and Fig 5?
3. The authors should add citation 21 (Pouliot, J. F. et al.) in the introduction as well, acknowledging that Glut1 Palmitoylation is already shown in BBB capillaries.
4. The authors should briefly explain the rationale, if any, for the selection of GBM cell lines, Ex: Is DHHC9 or Glut1 overexpressed in these cell lines?
5. There are scattered instances of stray articles ("the") or subject verb disagreement that should be addressed during editorial review, but overall the study is very well written.
6. The authors note that a multivariable analysis was performed, but I can find little detail on it in the methods section or in main text other than that age was included. Can additional details be provided?

What other relevant clinical factors were included? Performance status? Extent of resection? MGMT promoter methylation? Sex?

Reviewer #2 (Remarks to the Author); expert on palmitoylation and membrane proteins:

This study reports that S-palmitoylation of the glucose transporter GLUT1 promotes glucose uptake into tissues by stabilizing the plasma membrane localization of GLUT1. The authors map the site of palmitoylation of GLUT1 to a specific cysteine residue in the protein and demonstrate the DHHC9 is the enzyme responsible for mediating palmitoylation at this site. Using glioblastoma cell lines, they further show that glucose metabolism, cell growth, and tumorigenesis are dependent upon the palmitoylated cysteine of GLUT1 and on the catalytic activity of DHHC9. Relevance of these findings to human disease was established by showing that GLUT1 plasma localization and DHHC9 expression are positively correlated in human tumor specimens and is associated with a poor prognosis in human patients.

These findings are novel and biomedically significant. Although palmitoylation of GLUT1 has been reported, its functional significance has not been addressed in any biological context. The authors took a rigorous approach to establish the link between DHHC9 and GLUT1 and its pathophysiological impact. There are, however, a number of issues to be addressed to validate some of the experimental conclusions, and to improve the readability of the manuscript.

1. The scholarship of the manuscript would be improved by a more critical use of citations.

Line 62 – suggest citing the primary reference for the adipocyte palmitoylome rather than the review (which may be unrelated?)

Line 69 – cites a mix of 3 primary references and 3 reviews for a well established statement concerning the role of palmitoylation – the reviews are sufficient

Line 70 – cites reference 3 and 15 for information that is found in reference 22

Line 297 – two additional palmitoylation reviews were cited in the discussion in support of general statements about palmitoylation – why not use the earlier citations again?

2. The authors provide a convincing dataset that DHHC9 is a palmitoyltransferase for GLUT1. Given that there is substantial overlap of DHHC protein substrate specificity, it was somewhat surprising that a single DHHC protein came out of the CRISPR DHHC knockout screen (Ext. Data Fig. 2a). Was the knockout of each of the DHHCs confirmed by western blot with DHHC antibodies as suggested on line 506? If so, the source of DHHC antibodies should be provided. If the knockout was not confirmed, the authors should acknowledge that in the text and state that GLUT1 may be palmitoylated by other DHHC proteins.

3. Few studies in the field have confirmed the relationship between a substrate and its cognate DHHC protein using purified components. The authors are to be commended for this effort. However, it is not clear from the text or in the figures (e.g. there is no 16 kDa band apparent in the Coomassie-stained gel in Ext. Data Fig. 3d) whether DHHC9 was coexpressed with GCP16 to prepare a DHHC9/GCP16 complex or whether the proteins were expressed and purified separately and then reconstituted. Given the importance of GCP16 for DHHC9 enzyme stability and activity (ref. 22), additional information should be provided on the nature of the enzyme preparation. Figure 3b showing the in vitro data would be improved by also showing that DHHC9 is palmitoylated during the reaction.

4. The significant plasma membrane localization of endogenous DHHC9 in the GBM cell lines was somewhat surprising because several prior reports have indicated that DHHC9 is present predominately on intracellular membranes (e.g. Shimell JJ et al. Cell Rep. (2019) 29:2422-2437.) The commercial DHHC9 antibody was validated by the authors for western blots by the absence of signal in DHHC9 knockout cells. The authors should similarly validate that the signal is lost in immunofluorescence in the knockout cell line to insure the specificity of the DHHC9 antibody.

Minor points:

1. Molecular weight markers (MWM) are absent throughout the figures. Flanking MWM should be shown to document that the bands shown are the correct size, at least for the first figure where the proteins are shown.
2. Fig.1 legend – define GBM cells; “GBM cells (U87 or T98G cell lines).
3. Fig. 1 – It is evident from the immunofluorescence that non-palmitoylated Glut1 redistributes to intracellular membranes, rather than uniformly throughout the cytoplasm. Accordingly, in Fig. 1b and elsewhere, it might be better to change the label to Cyto/ICM and define the new label as cytoplasm/intracellular membranes in the figure legend. It would be nice to see a marker of intracellular membranes such as calnexin to validate that the plasma membrane fraction indeed represents a relatively pure PM fraction.
4. Fig.4a – the legend and figure aren't clearly informative as to what is measured. 2-DG uptake should be added to the y-axis label to indicate what luminescence represents.
5. Extended data Fig. 4b – Do the n=3 biologically independent samples represent tumor tissue from 3 different mice or from 3 tumor samples from a single mouse? Please clarify.

Reviewer #3 (Remarks to the Author); expert on metabolism:

Zhang et al. demonstrate that GLUT1 is palmitoylated at Cys207 by DHHC9, and this S-palmitoylation is required for maintaining GLUT1 at plasma membrane and promotion of glycolysis, cell proliferation, and brain tumorigenesis. In addition, DHHC9 expression positively are correlated with GLUT1 plasma membrane expression in GBM specimens and a poor prognosis in GBM patients.

The finding that GLUT1 palmitoylation by DHHC9 is novel and is of significance in understanding of the critical role of the regulation of glucose transporter in glycolysis and tumor development. The results are clean, robust, and supportive for the conclusion. The following points should be addressed before publication.

Points:

1. Both GLUT1 and GLUT3 are expressed in GBM. It is reported that GLUT3 expression levels are correlated with GBM poor prognosis and that the function of GLUT3 in glioblastoma cell is not recapitulated by GLUT1 (PMID: 33843470). Thus, it would be interesting to determine whether GLUT3 is S-palmitoylated by DHHC9 in GBM cells.
2. AKT activation promotes cell membrane expression of GLUT1. Is GLUT1 Cys207 palmitoylation dependent on AKT activation? Similarly, it has been reported that PKC-mediated GLUT1 S226 Phosphorylation is required for PM localization of GLUT1 (Molecular Cell, 2015). The author should test whether GLUT1 Cys207 palmitoylation is dependent on PKC activation.
3. The author should clarify whether the palmitoylation of GLUT1 occurs on the plasma membrane or cytosol.
4. Fig. 4A. 2-DG is not a natural substrate of GLUT1. Why was 2-DG uptake instead of glucose uptake measured although both of them can be uptaken by GLUT1?
5. Some sentences are not read well. A proof-reading of the manuscript would be helpful.

REVIEWER COMMENTS

Reviewer #1 (Remarks to the Author); expert on glioblastoma and metabolism:

This manuscript investigates how palmitoylation affects GLUT1 function and tumorigenesis in GBM. The study demonstrates that 90% of Glut1 is palmitoylated at C207, which promotes its localization at the plasma membrane. DHHC9 was identified as the palmitoyl acyltransferase responsible for Glut1 palmitoylation. Also, DHCC9 promotes growth, glycolysis and colony formation of GBM cells by palmitoylation of Glut1. A catalytically inactive mutant of DHHC9 or Glut1 C207S abrogates Glut1 palmitoylation of Glut1 and its PM distribution resulting in reduced growth, glycolysis and colony formation. Similar results were obtained in an orthotopic in vivo U87 GBM model. A positive correlation between DHCC9 expression and Glut1 PM localization was identified and is associated with poor prognosis in patient samples. Based on these results, the authors conclude that DHCC9 contributes to GBM tumorigenesis and DHHC9 as a promising target of GBM therapies. While there has been a prior report that GLUT1 palmitoylation might be important in brain capillaries, this is the first study to show such regulation in GBM. It further is the first study to describe the specific site of palmitoylation and the enzyme responsible for placing the palmitoyl group on GLUT1. Because of the novelty and potential therapeutic implications of this finding, my enthusiasm for this work is high. However, there are some issues detailed below that should be addressed.

Answer: We greatly appreciate the reviewer's acknowledgement of the potential significance of this report and the insightful comments, which are essential for the improvement of this manuscript. The necessary experiments have been performed to address the reviewer's outstanding and constructive questions.

Major issues:

1. The rigorous molecular biology studies are outstanding, and all phenotypes seem identical in the two immortalized GBM cell lines (U87, T98G) used throughout this work. However, immortalized GBM cell lines do not recapitulate much of the known biology of GBM human tumors and should not be the only models used if the authors wish to claim "DHHC9-mediated palmitoylation of GLUT1 is important in GBM". I would ask that the authors validate their cell line findings in a patient-derived model of GBM. Options for such a study could include patient-derived neurospheres, patient-derived adherent cells that have not been in culture for decades (<https://www.ncbi.nlm.nih.gov/pmc/articles/PMC4634360/>), or patient-derived xenografts. Due to the strength of rest of the paper and inclusion of robust human tumor data, I do not feel that additional in vivo animal work is needed, just a validation that deletion/knockdown of DHHC9 reduces GLUT1 PM localization in a better model than

U87 or T98G.

Answer: The reviewer's point is well taken. We depleted DHHC9 in a primary patient-derived GBM cell line PDX with adenoviruses encoding *DHHC9* short-hairpin RNA. Plasma membrane (PM) fraction and immunofluorescence analyses showed that GLUT1 PM localization decreased in PDX cells with DHHC9 depletion (**Supplementary Fig. 4d-f**).

2. I am not sure whether the regulation the authors describe is GBM-specific or also happens in normal brain. Because GLUT1 function is very important for normal brain (<https://academic.oup.com/brain/article/133/3/655/275024>), this distinction will be important for whether this regulation is a useful therapeutic strategy (i.e., it is GBM specific), or if it is simply a very interesting mechanistic finding. Studies regarding DHHC9-mediated GLUT1 palmitoylation in a non-transformed normal human astrocyte model could help address this issue.

Answer: We infected non-transformed Normal Human Astrocytes (NHA) cells with adenoviruses encoding *DHHC9* short-hairpin RNA. DHHC9 depletion markedly blocked the incorporation of alkynyl palmitic acid into GLUT1 (**Supplementary Fig. 5b**). Consistent results were obtained by using the APE assay, which showed that palmitoylation levels of GLUT1 were dramatically reduced by DHHC9 depletion (**Supplementary Fig. 5c**). In addition, subcellular fractionation analyses showed that GLUT1 was depleted from the PM fraction in DHHC9-depleted NHA cells (**Supplementary Fig. 5d**). These results suggest that DHHC9-mediated GLUT1 palmitoylation is required for GLUT1 PM localization in normal human astrocytes.

Minor issues:

1. I would suggest a slight re-wording of the title to: DHHC9-mediated GLUT1 S-palmitoylation promotes glioblastoma glycolysis and tumorigenesis

Answer: This point is well taken. We have used the title suggested by the reviewer in the revised manuscript.

2. Fig 4 and 5: Can the authors please add statistical significance (*) and the corresponding statistical tests with p-values in all the figure legends in Fig 4 and Fig 5?

Answer: We have added statistical significance (*) and the corresponding statistical tests with p-values in the figure legends of Fig 4 and Fig 5.

3. The authors should add citation 21 (Pouliot, J. F. et al.) in the introduction as well, acknowledging that Glut1 Palmitoylation is already shown in BBB capillaries.

Answer: We acknowledged that GLUT1 palmitoylation is already shown in BBB capillaries by adding the reference 7 (Pouliot, J. F. et al. PMID: 7696293) into the

Introduction section.

4. The authors should briefly explain the rationale, if any, for the selection of GBM cell lines, Ex: Is DHHC9 or Glut1 overexpressed in these cell lines?

Answer: We performed immunoblotting analyses and showed that expression of DHHC9 and GLUT1 were significantly upregulated in GBM cells comparing to NHA cells, which is normal human astrocytes (Supplementary Fig. 5a). Thus, we selected GBM cells to investigate the biological roles of DHHC9-mediated GLUT1 palmitoylation during tumorigenesis.

5. There are scattered instances of stray articles (“the”) or subject verb disagreement that should be addressed during editorial review, but overall the study is very well written.

Answer: We have corrected these mistakes in the revised manuscript.

6. The authors note that a multivariable analysis was performed, but I can find little detail on it in the methods section or in main text other than that age was included. Can additional details be provided? What other relevant clinical factors were included? Performance status? Extent of resection? MGMT promoter methylation? Sex?

Answer: The reviewer’s point is well taken. A multivariable analysis using clinical factors, including age, extent of resection, and sex has been performed (Supplementary Table 1). However, the performance status and MGMT promoter methylation, information of which are not available in the patient follow up database, were not included in this multivariable analysis.

Reviewer #2 (Remarks to the Author); expert on palmitoylation and membrane proteins:

This study reports that S-palmitoylation of the glucose transporter GLUT1 promotes glucose uptake into tissues by stabilizing the plasma membrane localization of GLUT1. The authors map the site of palmitoylation of GLUT1 to a specific cysteine residue in the protein and demonstrate the DHHC9 is the enzyme responsible for mediating palmitoylation at this site. Using glioblastoma cell lines, they further show that glucose metabolism, cell growth, and tumorigenesis are dependent upon the palmitoylated cysteine of GLUT1 and on the catalytic activity of DHHC9. Relevance of these findings to human disease was established by showing that GLUT1 plasma localization and DHHC9 expression are positively correlated in human tumor specimens and is associated with a poor prognosis in human patients.

These findings are novel and biomedically significant. Although palmitoylation of

GLUT1 has been reported, its functional significance has not been addressed in any biological context. The authors took a rigorous approach to establish the link between DHHC9 and GLUT1 and its pathophysiological impact. There are, however, a number of issues to be addressed to validate some of the experimental conclusions, and to improve the readability of the manuscript.

Answer: We greatly appreciate the reviewer's acknowledgement of the potential significance of this report and the insightful comments, which are essential for the improvement of this manuscript.

1. The scholarship of the manuscript would be improved by a more critical use of citations.

Line 62 – suggest citing the primary reference for the adipocyte palmitoylome rather than the review (which may be unrelated?)

Answer: The primary reference (PMID: 23599907) for the adipocyte palmitoylome has been cited to replace the previous one.

Line 69 – cites a mix of 3 primary references and 3 reviews for a well established statement concerning the role of palmitoylation – the reviews are sufficient

Answer: Three review articles were cited for this statement and the primary references have been removed in the revised manuscript.

Line 70 – cites reference 3 and 15 for information that is found in reference 22

Answer: To improve connection to the statement ahead, we have changed the original statement “DHHC9 forms a protein complex with GCP16 to function as a RAS PAT” to “For example, H-RAS and N-RAS are palmitoylated by DHHC9/GCP16 PAT complex, leading to plasma membrane (PM) localization of these proteins”. Original reference 3 (PMID: 22913968) has been removed, and original references 15 (PMID: 12193598) and 22 (PMID: 16000296) have been cited for this statement.

Line 297 – two additional palmitoylation reviews were cited in the discussion in support of general statements about palmitoylation – why not use the earlier citations again?

Answer: Three earlier citations (PMID: 25834228, PMID: 17892486, PMID: 17183362) have been used to replace these two additional palmitoylation reviews.

2. The authors provide a convincing dataset that DHHC9 is a palmitoyltransferase for GLUT1. Given that there is substantial overlap of DHHC protein substrate specificity, it was somewhat surprising that a single DHHC protein came out of the CRISPR DHHC knockout screen (Ext. Data Fig. 2a). Was the knockout of each of the DHHCs confirmed by western blot with DHHC antibodies as suggested on line 506? If so, the

source of DHHC antibodies should be provided. If the knockout was not confirmed, the authors should acknowledge that in the text and state that GLUT1 may be palmitoylated by other DHHC proteins.

Answer: The reviewer's point is well taken. We have added the statement "suggesting that DHHC9 is one of the PATs that regulate GLUT1 palmitoylation in GBM cells" on pages #7 and 8 of the manuscript, since the CRISPR/Cas9-mediated knockout of each DHHCs was not confirmed by western blot with DHHC antibodies.

3. Few studies in the field have confirmed the relationship between a substrate and its cognate DHHC protein using purified components. The authors are to be commended for this effort. However, it is not clear from the text or in the figures (e.g. there is no 16 kDa band apparent in the Coomassie-stained gel in Ext. Data Fig. 3d) whether DHHC9 was coexpressed with GCP16 to prepare a DHHC9/GCP16 complex or whether the proteins were expressed and purified separately and then reconstituted. Given the importance of GCP16 for DHHC9 enzyme stability and activity (ref. 22), additional information should be provided on the nature of the enzyme preparation. Figure 3b showing the *in vitro* data would be improved by also showing that DHHC9 is palmitoylated during the reaction.

Answer: The reviewer's point is well taken. The GST-tagged GCP16 and DHHC9 proteins were expressed and purified separately in sf9 insect cells and then reconstituted. The detailed experimental procedure is included in the Methods section. *Coomassie* Blue staining showed that purity of GST-tagged GCP16 and DHHC9 is more than 98% (Supplementary Fig. 4b). In addition, *in vitro* protein acyltransferase assay showed that wild-type (WT) DHHC9, but not a catalytically inactive DHHC9 C169S mutant, is able to incorporate palmitoyl alkyne in the presence of GCP16 (Supplementary Fig. 4b). These results suggested that we reconstituted an active palmitoyl acyltransferase (PAT) by mixing purified GST-tagged GCP16 and DHHC9 proteins.

We have performed the reviewer's suggested experiments. Fig. 3b shows that DHHC9 is also palmitoylated during the reaction.

4. The significant plasma membrane localization of endogenous DHHC9 in the GBM cell lines was somewhat surprising because several prior reports have indicated that DHHC9 is present predominately on intracellular membranes (e.g. Shimell JJ et al. Cell Rep. (2019) 29:2422-2437.) The commercial DHHC9 antibody was validated by the authors for western blots by the absence of signal in DHHC9 knockout cells. The authors should similarly validate that the signal is lost in immunofluorescence in the knockout cell line to insure the specificity of the DHHC9 antibody.

Answer: We performed the immunofluorescence staining in U87 cells using the commercial DHHC9 antibody (Thermo Fisher Scientific #PA5-56868). Immunofluorescence signal was lost in the *DHHC9*-knockout U87 cells (Supplementary Fig. 3e), suggesting that this antibody recognizes DHHC9 specifically.

Minor points:

1. Molecular weight markers (MWM) are absent throughout the figures. Flanking MWM should be shown to document that the bands shown are the correct size, at least for the first figure where the proteins are shown.

Answer: Molecular weight markers (MWM) have been labelled on the figures.

2. Fig.1 legend – define GBM cells; “GBM cells (U87 or T98G cell lines).

Answer: We have changed the “GBM cells” to “U87 or T98G cells” in the Fig.1 legend.

3. Fig. 1 – It is evident from the immunofluorescence that non-palmitoylated Glut1 redistributes to intracellular membranes, rather than uniformly throughout the cytoplasm. Accordingly, in Fig. 1b and elsewhere, it might be better to change the label to Cyto/ICM and define the new label as cytoplasm/intracellular membranes in the figure legend. It would be nice to see a marker of intracellular membranes such as calnexin to validate that the plasma membrane fraction indeed represents a relatively pure PM fraction.

Answer: The reviewer’s point is well taken. We extracted the intracellular membrane (ICM), which is a part of the cytoplasm, and plasma membrane (PM) proteins from U87 and T98G cells. Immunoblotting analyses indicated that the PM marker ATP1A1, but not the ICM marker calnexin and cytosolic marker GAPDH, was detected in the PM fractions (Fig.1b, 2f, 3g and Supplementary Fig. 4f, 5d). These results validate that the PM fractions are free of ICM and cytosolic proteins contamination.

We labelled the “whole cell lysate” as “WCL”, “intracellular membrane” as “ICM”, and “plasma membrane” as “PM” in the figures and accordingly defined these labels in the figure legends.

4. Fig.4a – the legend and figure aren’t clearly informative as to what is measured. 2-DG uptake should be added to the y-axis label to indicate what luminescence represents.

Answer: “2-DG uptake” was added to the y-axis labels of Fig. 4a.

5. Extended data Fig. 4b – Do the n=3 biologically independent samples represent tumor tissue from 3 different mice or from 3 tumor samples from a single mouse? Please clarify.

Answer: These samples represent tumor tissues from 3 different mice. A statement “Data were collected from n = 3 mice per group. The mean \pm s.d. values are shown.” was added to the legend of new Supplementary Fig. 6b.

Reviewer #3 (Remarks to the Author); expert on metabolism:

Zhang et al. demonstrate that GLUT1 is palmitoylated at Cys207 by DHHC9, and this S-palmitoylation is required for maintaining GLUT1 at plasma membrane and promotion of glycolysis, cell proliferation, and brain tumorigenesis. In addition, DHHC9 expression positively are correlated with GLUT1 plasma membrane lexpression in GBM specimens and a poor prognosis in GBM patients.

The finding that GLUT1 palmitoylation by DHHC9 is novel and is of significance in understanding of the critical role of the regulation of glucose transporter in glycolysis and tumor development. The results are clean, robust, and supportive for the conclusion. The following points should be addressed before publication.

Answer: We greatly appreciate the reviewer's insightful and positive comments, which significantly strengthened the manuscript.

Points:

1. Both GLUT1 and GLUT3 are expressed in GBM. It is reported that GLUT3 expression levels are correlated with GBM poor prognosis and that the function of GLUT3 in glioblastoma cell is not recapitulated by GLUT1 (PMID: 33843470). Thus, it would be interesting to determine whether GLUT3 is S-palmitoylated by DHHC9 in GBM cells.

Answer: To determine whether GLUT3 is palmitoylated, we performed a metabolic incorporation assay using a bioorthogonal fatty acid analog (alkynyl palmitic acid) with click chemistry conjugation. No obvious palmitoylated GLUT3 was detected (**Supplementary Fig. 1b**).

2. AKT activation promotes cell membrane expression of GLUT1. Is GLUT1 Cys207 palmitoylation dependent on AKT activation? Similarly, it has been reported that PKC-mediated GLUT1 S226 Phosphorylation is required for PM localization of GLUT1 (Molecular Cell, 2015). The author should test whether GLUT1 Cys207 palmitoylation is dependent on PKC activation.

Answer: We performed the reviewer's suggested experiments. The GLUT1 palmitoylation is not obviously altered by treating U87 cells with PI3K inhibitor LY294002 or PKC inhibitor Gö 6983, which suppresses AKT and PKC activation, respectively (**Supplementary Fig. 5e**), suggesting that DHHC9-mediated GLUT1 palmitoylation is independent of AKT and PKC activation.

3. The author should clarify whether the palmitoylation of GLUT1 occurs on the plasma membrane or cytosol.

Answer: The point is well taken. IF analyses with specificity-validated antibodies (**Supplementary Fig. 3e**) demonstrated that endogenous DHHC9 and GLUT1 colocalized at the PM of U87 and T98 cells (**Supplementary Fig. 3f**). Consistently, robust reconstituted GFP fluorescence was observed at the PM of U87 and T98 cells in

a split-GFP system coexpressing C-terminal split-GFP (S1-10)-tagged DHHC9 and split-GFP (S11)-tagged GLUT1 (Supplementary Fig. 3g, h), demonstrating that DHHC9 and GLUT1 colocalized at the PM of these cells. These results suggest that DHHC9 palmitoylates GLUT1 at the PM of GBM cells.

4. Fig. 4A. 2-DG is not a natural substrate of GLUT1. Why was 2-DG uptake instead of glucose uptake measured although both of them can be uptaken by GLUT1?

Answer: Molecular structure of 2-deoxyglucose (2-DG) is similar to glucose. 2-DG taken up by glucose transporters is metabolized to 2-DG-6-phosphate (2-DG6P), which cannot be further metabolized, thereby accumulates within cells. The level of accumulated 2-DG6P determined by an enzymatic reaction coupling to NADPH generation is directly proportional to 2-DG (or glucose) uptake by cells. Thus, the activity of glucose transporters, for example GLUT1, was determined by 2-DG uptake rather than by glucose uptake.

5. Some sentences are not read well. A proof-reading of the manuscript would be helpful.

Answer: We have proofread the revised manuscript.

REVIEWER COMMENTS

Reviewer #1 (Remarks to the Author):

My comments have largely been addressed and the revised manuscript is significantly improved. The observation that DHHC9 appears to regulate GLUT1 plasma membrane expression in non-malignant cells (NHA) in addition to GBM makes its inhibition less likely to have selectivity for cancer cells compared to normal. I would like the authors to discuss this finding, and its implication for therapy/selectivity, in the discussion.

I also have lingering questions about the multivariate analysis. "Total Resection" is listed as a co-variate twice in supplementary Table 1 and has a different hazard ratio each time it is listed. I am not sure how this can be correct. The word "regression" also seems like it is cut off in the title. Lastly, I am not sure how "high" vs. "low" was defined for the protein levels (DHHC9 expression and PM-localized GLUT1 expression). Was it a cut at the median? Or were certain IHC scored deemed "high" and others deemed "low". I would ask that the authors add these details to the methods.

Reviewer #2 (Remarks to the Author):

This study reports that S-palmitoylation of the glucose transporter GLUT1 promotes glucose uptake into tissues by stabilizing the plasma membrane localization of GLUT1. The authors map the site of palmitoylation of GLUT1 to a specific cysteine residue in the protein and demonstrate the DHHC9 is the enzyme responsible for mediating palmitoylation at this site. Using glioblastoma cell lines, they further show that glucose metabolism, cell growth, and tumorigenesis are dependent upon the palmitoylated cysteine of GLUT1 and on the catalytic activity of DHHC9. Relevance of these findings to human disease was established by showing that GLUT1 plasma localization and DHHC9 expression are positively correlated in human tumor specimens and is associated with a poor prognosis in human patients.

These findings are novel and biomedically significant. Although palmitoylation of GLUT1 has been reported, its functional significance has not been addressed in any biological consequence. The authors took a rigorous approach to establish the link between DHHC9 and GLUT1 and its pathophysiological impact.

The authors have satisfactorily addressed the concerns that were raised in the initial review.

Reviewer #3 (Remarks to the Author):

The authors have successfully addressed the reviewer's previous concerns.

REVIEWER COMMENTS

Reviewer #1 (Remarks to the Author):

My comments have largely been addressed and the revised manuscript is significantly improved. The observation that DHHC9 appears to regulate GLUT1 plasma membrane expression in non-malignant cells (NHA) in addition to GBM makes its inhibition less likely to have selectivity for cancer cells compared to normal. I would like the authors to discuss this finding, and its implication for therapy/selectivity, in the discussion.

Answer: We appreciate the reviewer's insightful comments again and this point is well taken. To discuss our finding that DHHC9 also regulates plasma membrane localization of GLUT1 in NHA cells in addition to GBM cells and its implication for therapy/selectivity, we have added a paragraph, which is highlighted in yellow, into the Discussion section of the revised manuscript.

Moreover, to support this discussion, we have performed extra experiments and found that DHHC9 depletion did not significantly alter glucose uptake or the glycolytic rate in NHA cells (Supplementary Fig. 6a-c), which have lower expression of DHHC9 compared to the GBM cells (Supplementary Fig. 5a).

I also have lingering questions about the multivariate analysis. "Total Resection" is listed as a co-variate twice in supplementary Table 1 and has a different hazard ratio each time it is listed. I am not sure how this can be correct.

Answer: In Supplementary Table 1, two different multivariate analyses of overall survival (OS) were performed by using one set of 4 prognostic factors, including age, sex, total resection, and DHHC9 staining scores, and another set of 4 prognostic factors, including age, sex, total resection, and PM GLUT1 staining scores, to estimate the survival risk of each prognostic factor in the cohort of GBM patients. The hazard ratio (HR) values for each prognostic factor were calculated by using the Cox's regression model, and these values could change in multivariate OS analyses containing different prognostic factors, therefore the total resection's HR value obtained from the DHHC9 multivariate OS analysis could be different to that obtained from the PM GLUT1 multivariate OS analysis.

The word "regression" also seems like it is cut off in the title.

Answer: We are sorry for the mistake caused by converting the Supplementary Table 1 from Excel format to PDF format. The correct title of Supplementary Table 1 is "Multivariate analysis of overall survival in GBM patients (Cox's regression model)". We have corrected this mistake in the revised manuscript.

Lastly, I am not sure how "high" vs. "low" was defined for the protein levels (DHHC9 expression and PM-localized GLUT1 expression). Was it a cut at the median? Or were certain IHC scored deemed "high" and others deemed "low". I would ask that the authors add these details to the methods.

Answer: We defined the tissue sections with staining scores of 0 to 4 and 5 to 8 as low staining and high staining, respectively. These details have been added into the Method section in the revised manuscript.

REVIEWER COMMENTS

Reviewer #1 (Remarks to the Author):

My comments have been satisfactorily addressed. Thank you.